# Targeting TAZ-TEAD in minimal residual disease enhances the duration of targeted therapy in melanoma models

Connor A. Ott[1], Timothy J. Purwin [1], Pan-Yu Chen[2], Somenath Chowdhury[2], George L. Mellor[1], Kristine Luo [1], Glenn L. Mersky [1], Manoela Tiago[1], William D. Madden[1], Scott D. Varney[1], Dan A. Erkes[1], John Lamar [3], Claudia Capparelli [4,5], Gideon Bollag[2] & Andrew E. Aplin [1,5] ✉

Targeted therapies in cancer are limited by cells exhibiting drug tolerance. We aimed to target drug tolerance in order to delay the development of acquired resistance. In melanoma, tolerance to MAPK pathway inhibitors is associated with loss of SOX10 and an enhanced TEAD transcriptional program. We show that loss of SOX10 is sufficient to up-regulate TEAD targets with dependence on the co-activator, TAZ. Active TAZ is sufficient to mediate tolerance to BRAF inhibitors and MEK inhibitors. We develop two covalent inhibitors, OPN-9643 and OPN-9652, designed to target the central palmitate binding pocket of TEADs. In SOX10-deficient cells, OPN-9643 and OPN-9652 reduce TEAD-dependent reporter activity and expression of TEAD targets, CTGF and CYR61. OPN-9643 and OPN-9652 treatment enhances the inhibitory effects of MAPK-targeted therapies in 2D and 3D growth assays in SOX10 knockout cells and reverses tolerance mediated by active TAZ. In vivo, OPN-9652 delays the onset of acquired resistance to BRAF inhibitors and MEK inhibitors from minimal residual disease. Thus, TAZ-TEAD activity plays an important role in melanoma drug tolerance and the development of acquired resistance.

Targeted therapies in cancer are limited by acquired resistance. A prevailing concept is that drug-tolerant persisters constitute a reservoir of surviving cells from which fully drug-resistant cells emerge and seed tumor regrowth[1–3]. Since melanomas are plastic and frequently progress on targeted therapy treatment, they represent a compelling model for studying drug-tolerant persisters[4]. BRAF V600E mutations are present in approximately 50% of melanomas and hyperactivate the MEK-ERK1/2 signaling pathway[5,6]. BRAF inhibitors (BRAFi) and MEK inhibitors (MEKi) are approved to treat late-stage BRAF V600E melanomas[7], but most tumors acquire drug resistance with patients having a median progression-free survival of 12-14 months[8–10]. Heterogeneous mechanisms of resistance have been identified[11]. Analysis of drug-tolerant tumor patient samples identifies four distinct drug-tolerant states following BRAFi and MEKi: i) invasive; ii) neural crest stem cell-like (NCSC); iii) pigmented, and iv) "starved"- like melanoma cells (SMC)[12]. Further analysis indicates that SOX10 expression is lost in a subpopulation of invasive minimal residual disease (MRD) cells following BRAFi + MEKi treatment[12] and in acquired resistant tumors[13]. The invasive state is characterized by low expression of SOX10, high expression of AXL, and an innate tolerance to BRAFi and MEKi in vitro[14,15]. SOX10 is a melanocytic lineage-specific transcription factor that is highly but heterogeneously expressed in melanoma. SOX10 expression promotes proliferation and tumor growth[16,17], whereas its loss is sufficient to induce an invasive, slow-cycling state[18,19]. SOX10-

[1]Department of Pharmacology, Physiology, and Cancer Biology, Thomas Jefferson University, Philadelphia, PA 19107, USA. [2]OpnaBio LLC, South San Francisco, CA, USA. [3]Department of Molecular and Cellular Physiology, Albany Medical College, Albany, NY, USA. [4]Department of Medical Oncology, Thomas Jefferson University, Philadelphia, PA, USA. [5]Sidney Kimmel Comprehensive Cancer Center at Jefferson Health, Philadelphia, PA, USA. ✉e-mail: Andrew.Aplin@Jefferson.edu

deficient cells are intrinsically tolerant to BRAFi and/or MEKi[13,18]; however, targetable mechanisms of drug tolerance remain unclear.

One pathway linked to drug tolerance is Hippo signaling mediated by the MST1/2 and LATS1/2 kinase cascade and their respective cofactors, SAV1 and MOB1[20]. LATS1/2 phosphorylate the transcriptional coactivators, Yes-associated protein 1 (YAP1) and WW-domain containing transcriptional regulator 1 (WWTR1), also known as TAZ[21–23]. Phosphorylation of YAP1 and TAZ promotes both their cytoplasmic sequestration through 14-3-3 binding and proteasomal degradation[24,25]. When YAP1 and TAZ are dephosphorylated at multiple serine residues, they translocate into the nucleus and interact with the TEAD family of transcription factors[26,27]. AP-1 family proteins can cooperate with TEADs to regulate their transcriptional output by recruiting TEAD transcription factors to target downstream gene targets[28–30]. The YAP1/TAZ-TEAD pathway is altered in cancers. For example, mutations in *NF2*, an upstream regulator of the Hippo pathway, occur in >40% of malignant pleural mesothelioma cases, and gene fusions in YAP1 and TAZ have been identified[31]. YAP1 and TAZ have been associated with resistance towards targeted, chemo- and radiotherapies in lung and breast cancers[32–36]. Increased YAP1 activity is detected in drug-resistant patient tumors[37] and promotes resistance to BRAFi in melanoma, colon, and thyroid cancer[32].

In this study, the goal was to target drug tolerance to BRAFi and/or MEKi in melanoma in order to reduce the onset of acquired resistance. We show that SOX10 loss is sufficient to up-regulate the TEAD pathway through a TAZ-dependent mechanism. We generate TEAD inhibitors that resensitize SOX10-negative, drug-tolerant cells to BRAFi + MEKi. TEAD inhibitors delay the onset of acquired resistance to BRAFi + MEKi. Together, our findings provide a strategy to target melanoma before they acquire resistance to targeted therapies with the aim of enhancing durable effects in the clinic.

## Results

### YAP1/TAZ signaling is up-regulated in SOX10-deficient cells

SOX10-deficient melanoma cells show invasive features and tolerance to BRAFi and/or MEKi[18]. Additionally, SOX10 loss is associated with resistance to immune checkpoint inhibitors[38]. We utilized two melanoma cell models: A375 cells (BRAF V600E), which are MITF-low, neural crest stem cell (NCSC)-like cells[39]; and MeWo cells (BRAF wild-type), which express the melanocytic cell state markers SOX10 and MITF[18]. Gene set enrichment analysis (GSEA) of A375 SOX10 knockout (KO) cells and MeWo SOX10 KO cells versus their respective parental cells showed positive enrichment of a mesenchymal-like gene signature (Fig. 1A), which is characterized by high expression of AXL[14,15]. In the Rambow et al. single-cell RNA-seq dataset[12], the MRD-characterized invasive state showed low levels of SOX10 and elevated levels of two canonical TEAD targets, CTGF and CYR61 (Fig. 1B). We further analyzed expression of other cell state markers and observed that expression of MITF and MelanA, a marker of the melanocytic cell state[40], were significantly lower in the drug-tolerant invasive cell state compared to other cell states. Conversely, KDM5B expression, which has been linked to a slow-cycling and drug-tolerant phenotype[41–43], was higher in the invasive cell state (Supplementary Fig. 1A). Consistently, our lab has previously shown that MeWo SOX10 KO cells lose expression of MITF following SOX10 depletion[18]. MelanA expression was reduced in SOX10 KO in MeWo and not expressed in A375 cells (Supplementary Fig. 1B). NGFR was only expressed in the NCSC-like A375 cells[39], and was decreased in A375 SOX10 KO cells (Supplementary Fig. 1B). This is consistent with NGFR expression in the SOX10-expressing/MITF-low NCSC state[12,44].

Other studies have shown upregulated TEAD signaling in the SOX10-low undifferentiated/invasive/mesenchymal cell state[1,45]. To directly test whether SOX10 loss is sufficient to up-regulate YAP1/TAZ-TEAD signature genes, we compared RNA-seq data from SOX10 KO cells to previously published YAP1/TAZ gene sets up-regulated

following YAP1 overexpression. Publicly available bulk RNA-seq datasets from Capparelli et al.[18]. showed a positive enrichment of three available YAP-TAZ gene signatures (YAP/TAZ melanoma up[46], YAP/TAZ cancer up, and Harvey Melanoma Up[47]) in both A375 and MeWo SOX10 KO cell lines compared to parental cells (Fig. 1C, Supplementary Fig. 1C). Up-regulation of TEAD targets in SOX10 KO A375 and MeWo cells was further validated through Western blot analysis for CTGF and CYR61 (Fig. 1D).

Mechanistically, how SOX10 loss leads to up-regulation of TEAD targets is unknown. Mutations in the *NF2* gene, which encodes for the tumor suppressor protein, Merlin, give rise to dysregulation in the Hippo pathway and increased YAP1/TAZ nuclear translocation[48]. Merlin expression was decreased in A375 SOX10 KO compared to parental cells (Fig. 1E). By contrast, no alteration in Merlin expression was detected in MeWo SOX10 KO models. We detected an increase in YAP1, TEAD1, and pan-TEAD expression in A375 SOX10 KO cells compared to parental, but there were no changes in expression of TAZ or other TEAD paralogs (Supplementary Fig. 1D, E). YAP1 phosphorylation at S127, S109, and S61 increased concomitant with YAP1 expression increases in A375 SOX10 KO cells (Supplementary Fig. 1D). In contrast, there were no changes in either YAP1 phosphorylation at S127 and S61 or in total YAP1, TAZ, or TEAD expression between MeWo parentals and MeWo SOX10 KO cells; however, there was a decrease in pYAP1 S109. (Supplementary Fig. 1D, E).

Given the role of AP-1, as well as TEADs, as regulators of the invasive cell state[45], we analyzed an AP-1 gene signature and found an enrichment in both A375 and MeWo SOX10 KO cells (Supplementary Fig. 2A). Previous studies indicated that SOX10-deficiency induces high chromatin remodelling[39]. Consistently, analysis of publicly available ATAC-seq data[39] revealed that loss of SOX10 induces an enrichment in TEAD binding motifs, indicating that epigenetic mechanisms also regulate TEAD binding in SOX10-deficient cells (Supplementary Fig. 2B). Further analysis of publicly available ATAC-seq datasets revealed increased chromatin accessibility at the transcription start sites of CTGF and CYR61 in SOX10-negative cells (MM029, MM099, and MM047) compared to SOX10-proficient cells (Supplementary Fig. 2C). These findings suggest that loss of SOX10 is associated with an open chromatin state at these loci, potentially facilitating transcriptional upregulation of CTGF and CYR61. Furthermore, analysis of RNA-seq data revealed a statistically significant upregulation of c-Jun expression in A375 and MeWo SOX10 KO cell lines (Supplementary Fig. 2D), which was further validated by western blot (Supplementary Fig. 2E). Depletion of c-Jun through siRNA knockdown in A375 SOX10 KO cell lines showed a reduction in CTGF and CYR61 expression (Supplementary Fig. 2F). Overall, these data show that despite no overt change in TAZ/TEAD expression, SOX10-low, drug-tolerant cells exhibit up-regulation of YAP1/TAZ-TEAD signaling in melanoma, likely through multiple mechanisms including chromatin remodeling and c-Jun upregulation.

### TAZ is a major regulator of TEAD targets in SOX10-deficient melanoma cells

YAP1 and TAZ have partially overlapping functions as coactivators of TEADs[49]. We examined the dependency on YAP1 and TAZ expression of canonical TEAD targets in SOX10-deficient melanoma by siRNA knockdown. TAZ depletion significantly reduced CTGF and CYR61 expression, while YAP1 depletion elicited minor effects in A375 and MeWo SOX10 KO cells (Fig. 2A). To characterize the YAP1- and TAZ-regulated transcriptomes in SOX10-lowdrug-tolerant melanoma, RNA-seq and GSEA were performed following depletion of either YAP1 or TAZ in parental and SOX10 KO A375 cells. RNA-seq analysis confirmed a reduction in YAP1 and TAZ expression after siRNA knockdown (Fig. 2B). Principal component analysis (PCA) did not separate YAP1 knockdown and TAZ knockdown in A375 parental cells from control A375 cells, likely since the downstream activity is already low in the

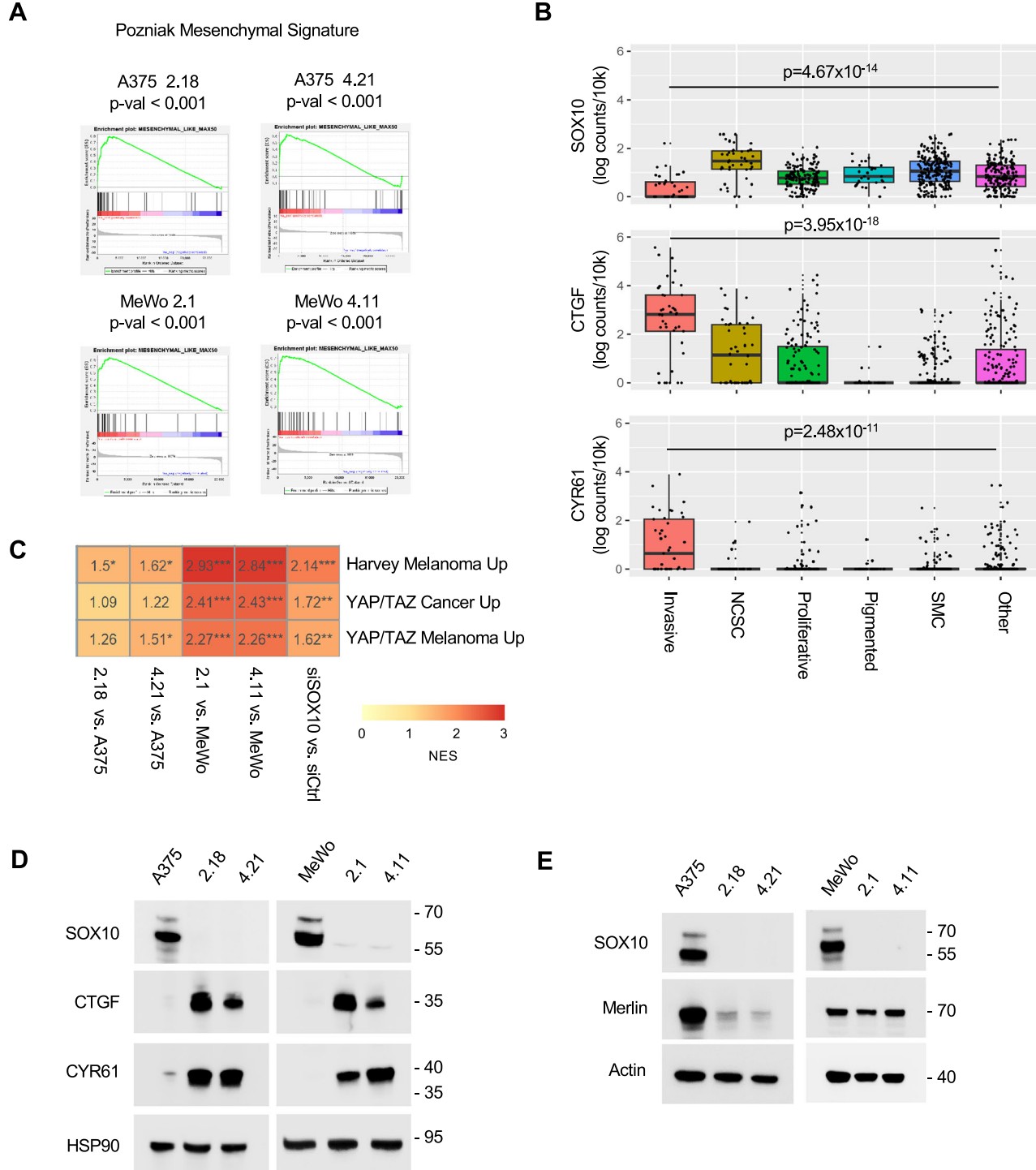

**Fig. 1 | YAP1/TAZ-TEAD signaling is up-regulated in SOX10-deficient cells.**
**A** Enrichment plots of GSEA results showing up-regulation of a mesenchymal gene signature[38] for A375 and MeWo CRISPR SOX10 KO cells vs parental cells. ***p < 0.001, two-sided permutation test. **B** Box plots of SOX10, CTGF, and CYR61 expression levels by invasive (n = 41), NCSC (n = 44), proliferative (n = 147), pigmented (n = 30), SMC (n = 224) and other (n = 188) cell states in a scRNA-seq dataset of patient-derived xenograft melanomas following BRAFi + MEKi from Rambow et al.[12]. The Seurat FindMarkers() function was used with the two-sided Likelihood-ratio test for single cell gene expression[111] to determine differentially expressed genes between the invasive and all other cell states. SOX10 log2FC = 1.96, CTGF log2FC = 2.94, CYR61 log2FC = −0.894. Box plots are made with ggplot2::geom_boxplot() using default summary statistic parameters, which show the 25% quantile,

median, and 75% quantile for the lower bound, center line, and upper bound of the box, respectively. Box plot whiskers are drawn to the lowest or highest data point within 1.5 * IQR from the lower or upper bound of the box for the lower and upper whiskers, respectively. **C** Heatmap showing normalized enrichment scores (NES) of GSEA YAP1/TAZ gene signatures comparing parental A375 and MeWo cells to SOX10 KO cell lines and an independent dataset with A375 SOX10 knockdown samples. *p < 0.05, **p < 0.01, ***p < 0.001, two-sided BHFDR. **D** A375 parental, A375 2.18 SOX10 KO, A375 4.21 SOX10 KO cell lysates and MeWo parental, MeWo 2.1 SOX10 KO, and MeWo 4.11 SOX10 KO cell lysates were analyzed by Western blotting with the antibodies to SOX10, CTGF, CYR61, and HSP90 (loading control). The experiment was repeated independently three times with similar results. **E** As above, with Western blotting for SOX10, Merlin and actin.

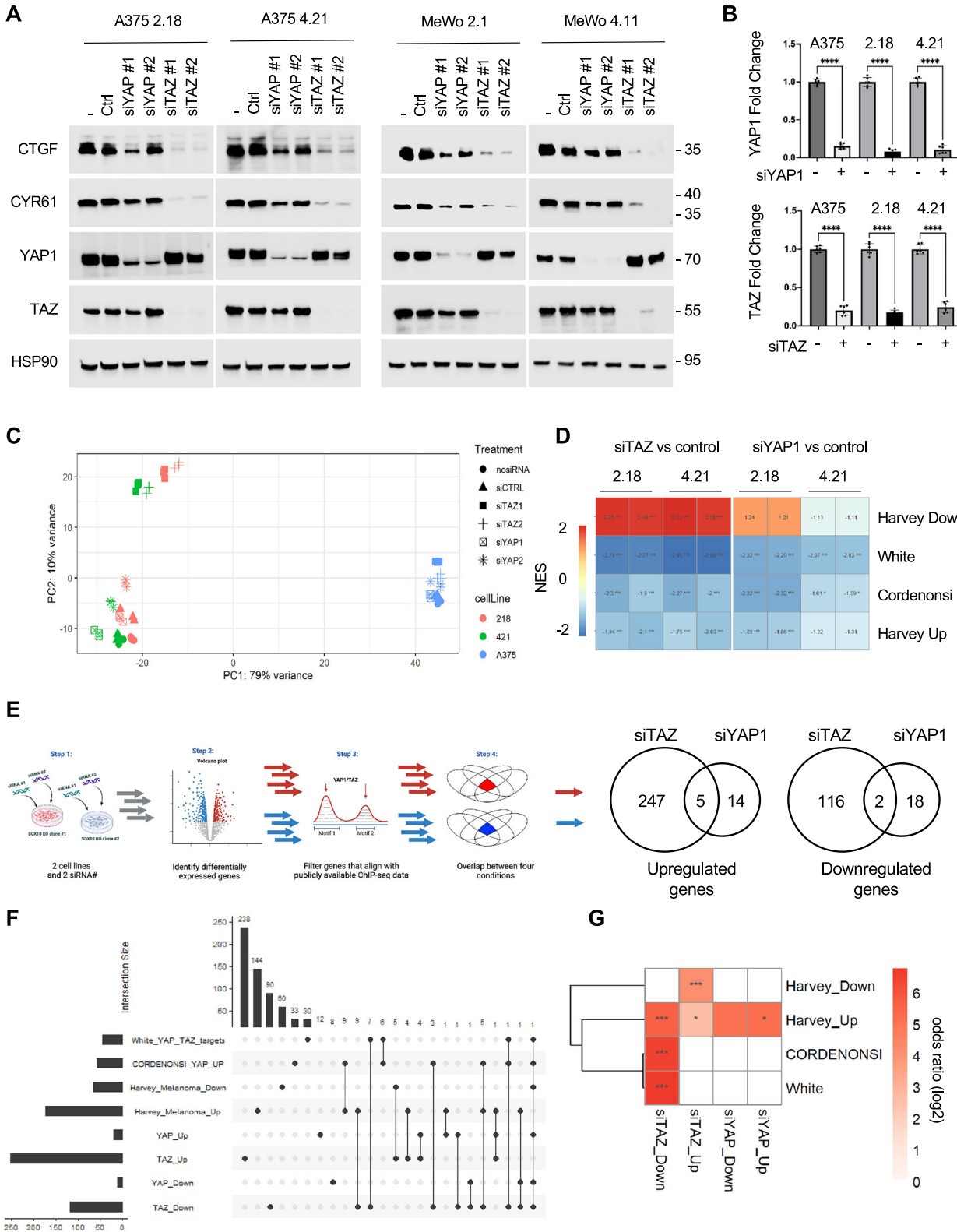

parental cells (Fig. 2C). These data indicate that YAP1 and TAZ depletion do not elicit major transcriptomic changes in SOX10-expressing cells. By contrast, in SOX10 KO cells, which display high YAP1/TAZ-TEAD signaling, we detected distinct clustering of TAZ knockdown samples away from a cluster of YAP1 knockdown samples and control samples. These results indicate that in melanoma cells with high TEAD activity, TAZ depletion has a large transcriptomic effect, which is not detected in YAP1 knockdown cells.

Next, we compared our RNA-seq data to four previously established YAP1/TAZ gene signatures (Cordenonsi YAP UP[50], White YAP/TAZ[51], Harvey Melanoma Up, and Harvey Melanoma Down[47]). We observed a greater enrichment in all 4 gene sets following TAZ knockdown compared to YAP1 knockdown (Fig. 2D), suggesting that TAZ is responsible in melanoma cells depleted of SOX10 for mediating the expression of genes associated with these YAP1/TAZ signatures. HALLMARK gene sets were analyzed in A375 SOX10 KO cells, and we

**Fig. 2 | YAP1 and TAZ regulate distinct transcriptomes. A** A375 crSOX10 #2.18, A375 crSOX10 #4.21, MeWo crSOX10 #2.1, and MeWo crSOX10 #4.11 cells were untreated or treated with reagent alone, non-targeting control siRNA, siYAP1, or siTAZ. After 72 hrs, cells were lysed and lysates were analyzed by Western blotting with the antibodies indicated. The experiment was repeated independently three times with similar results. **B** Barplot of fold change values of RNA-seq of YAP1 and TAZ following knockdown, Data are expressed as mean ± SD. n = 6. ****p < 0.0001 one-way ANOVA. Source Data are available. **C** PCA plot of A375 parental, 2.18 SOX10 KO cells, and 4.21 SOX10 KO cells treated with transfection reagent alone (-), non-targeting control, YAP1, or TAZ siRNA. **D** Heatmap showing NES of YAP1/TAZ gene signatures following knockdown of either YAP1 or TAZ in SOX10 KO cell lines compared to control cells. *p < 0.05, **p < 0.01, ***p < 0.001 two-sided BHFDR. **E** Visual schematic outlining process to produce gene signature for YAP1 and TAZ and Venn diagram of gene signatures. The numbers of genes that are significantly regulated by either YAP1, TAZ, or both are indicated. Created in BioRender. Ott, C. (2025) https://BioRender.com/d0vmn2m. **F** Upset plot showing the overlap of members between 4 published YAP1/TAZ signatures[47,50,51] and 4 YAP1/TAZ signatures generated in this study (siYAP1_Up, siTAZ_Up, siYAP1_Down, siTAZ_Down). **G** Heatmap of log2-transformed odds ratio values for the overlap of publicly available YAP1 and TAZ gene signatures in the refined YAP1 and TAZ signatures generated in this study. *p < 0.05, **p < 0.01, ***p < 0.001 two-sided Fischer's exact test with BHFDR multiple comparison adjustment.

observed a positive enrichment in several pathways (including apical surface, myogenesis, UV response down, apoptosis, apical junction, hedgehog signaling, and IL2-STAT5 signaling) following TAZ knockdown that were negatively enriched by YAP1 knockdown (Supplementary Fig. 3A). Conversely, several pathways (MYC targets, unfolded protein response, E2F targets, G2M checkpoint, oxidative phosphorylation, and DNA repair) were negatively enriched in TAZ knockdown samples that were positively enriched in YAP1 knockdown samples (Supplementary Fig. 3A). We analyzed the top five positively and negatively enriched pathways in TAZ and YAP1 knockdown samples. Only KRAS signaling in the positively enriched groups and MTORC1 signaling in the negatively enriched groups were pathways shared between TAZ and YAP1 knockdowns (Supplementary Fig. 3B). To determine the degree of enrichment between siTAZ and siYAP1 samples in each gene set, we plotted the absolute value of the NES between TAZ knockdown and YAP1 knockdown samples. The median absolute NES score for siTAZ gene sets was 1.59, and only 0.81 in siYAP1 gene sets, indicating that TAZ knockdown resulted in a greater degree of enrichment when compared to YAP1 (Supplementary Fig. 3C).

We developed our own unique signatures based on genes that were up-regulated or down-regulated following knockdown of either YAP1 or TAZ and further filtered genes based on targets within YAP1 and TAZ ChIP-seq data[52] to create up-regulated and down-regulated gene signatures for YAP1 and TAZ (Fig. 2E, Supplementary Data 1). Significantly more genes were either up-regulated or down-regulated following TAZ knockdown compared to YAP1 knockdown (Fig. 2E), indicating that TAZ has a major role in regulating gene expression in SOX10 KO melanoma cells. We compared the overlap of our signature to other established YAP1/TAZ signatures and detected 26 shared genes between the TAZ_down signature but only 1 shared gene in the YAP1_down signature (Fig. 2F). TAZ knockdown gene signatures showed significant concordance with the four aforementioned publicly available YAP/TAZ gene signatures (Fig. 2G). Overall, these data suggest that TAZ is the major co-activator of the TEAD transcriptome in SOX10-low cutaneous melanoma cells.

## Melanoma cells have a greater predicted dependence on TAZ than YAP1

To further examine co-activator selective effects, we utilized Chronos CRISPR gene dependency score data from the Cancer Dependency Map (DepMap) to analyze the TAZ and/or YAP1 requirement across 62 melanoma cell lines[53]. Predicted dependency is indicated by a Chronos score below -1. This dataset reflects gene dependency in drug-naïve, basal culture conditions and we identified five (out of 62) melanoma cell lines with a predicted dependency on *WWTR1*/TAZ (Fig. 3A). No cell lines within DepMap had a predicted dependency on YAP1. Overall, the mean dependency score in the 62 cell lines was significantly lower for TAZ than YAP1 (Fig. 3B). Furthermore, in drug-naïve basal conditions, there was no correlation between SOX10 expression and predicted YAP1 or TAZ dependency (Supplementary Fig. 4A); however, there was a correlation between predicted TAZ dependency and predicted TEAD1 dependency (Fig. 3C). We tested WM983B cells, one of

the predicted TAZ-dependent melanoma cell lines. Knockdown of TAZ alone in WM983B cells led to a dramatic down-regulation of CTGF and CYR61 expression, whereas YAP1 knockdown resulted in little to no reduction (Fig. 3D). We did not observe any changes in the expression of SOX10 following the knockdown of either TAZ or YAP1. Furthermore, WM983B cell growth was significantly reduced by TAZ knockdown (p < 0.0001) compared to siControl-treated cells (Fig. 3E). YAP1 knockdown elicited no or modest effects on cell growth depending on the siRNA sequence utilized. Overall, these data suggest that while TAZ is the primary TEAD co-activator in melanoma, its requirement for cell growth is occasionally observed in melanoma cell lines in the absence of targeted therapy.

To test whether active TAZ is sufficient to induce the expression of CYR61 and CTGF, we utilized A375 parental cells, which exhibit a low level of TEAD activity. We constructed a doxycycline-inducible TAZ containing a point mutation, S89A, that prevents phosphorylation and is insensitive to LATS-mediated cytoplasmic sequestration[54,55] and compared its effects to YAP1 containing the equivalent S127A mutation. Expression of CTGF was up-regulated effectively by induction of TAZ-S89A, and a modest increase in CYR61 expression was detected in A375 cells (Fig. 3F) and MeWo cells (Supplementary Fig. 4B). Immunofluorescence for HA-tag in the exogenous TAZ-S89A cells confirmed that expressed TAZ is localized in the nucleus (Fig. 3G). No effect on CTGF and CYR61 expression was observed following the induction of LacZ. Despite the utilization of TAZ, we did not detect a change in TAZ nuclear localization via immunofluorescence analysis comparing A375 parental and SOX10 KO cells (Supplementary Fig. 4C), although we note that immunofluorescence cannot rule out that small changes in TAZ localization occur that may impact the downstream transcriptional output. While no correlation between SOX10 expression and TAZ dependency is observed in the drug naïve state, further analysis is warranted to determine the mechanisms by which TAZ drives a SOX10-deficient drug-tolerant phenotype. Overall, these data suggest a more substantial role for TAZ versus YAP1 in regulating the transcriptome in melanoma cells.

## TAZ is sufficient to promote drug tolerance to BRAFi + MEKi through TEAD

Since YAP1 and TAZ elicited differential effects on transcriptomes, we examined the dependency of YAP1 and TAZ on drug resistance. We determined whether TAZ-S89A was sufficient to induce drug tolerance as measured by increased growth in the presence of BRAFi + MEKi using IncuCyte assays[56]. When A375 parental cells were cultured in the presence of BRAFi + MEKi, induction of TAZ-S89A enhanced growth in the presence of BRAFi + MEKi although did not rescue growth to levels observed in the absence of BRAFi + MEKi (Fig. 4A). YAP1-S127A induction also increased cell growth in BRAFi + MEKi treatment conditions, while LacZ elicited no effect. Similarly, induction of either TAZ-S89A or YAP-S127A in MeWo cells promoted growth in MEKi conditions (Supplementary Fig. 4D).

To test whether TAZ knockdown was sufficient to reduce the growth following BRAFi + MEKi in A375 SOX10 KO cells, we knocked down TAZ in the absence/presence of BRAFi + MEKi. Knockdown of

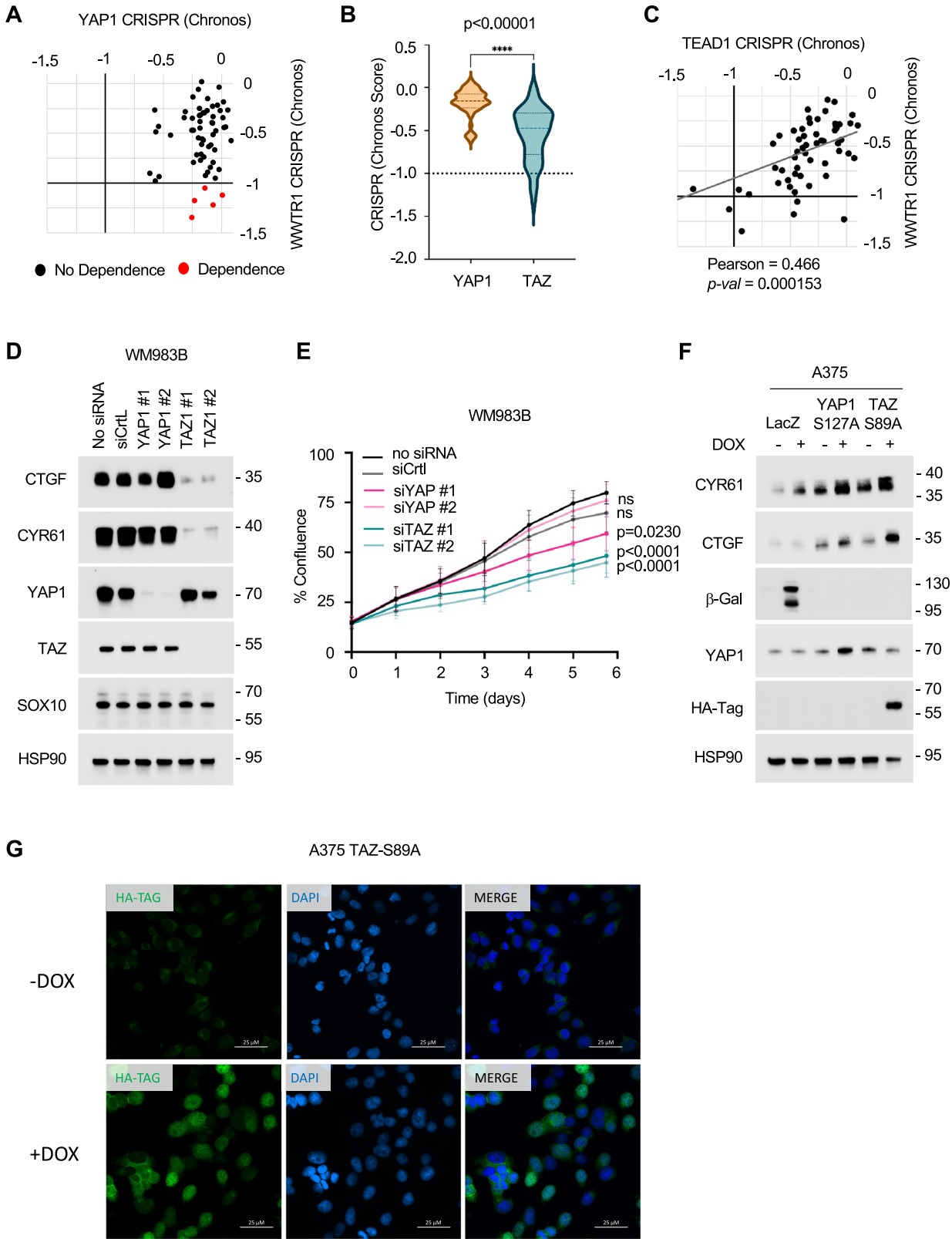

TAZ resensitized 2.18 and 4.21 SOX10 KO cells to BRAFi + MEKi, resulting in a significant decrease in cell growth. (Fig. 4B). No significant growth effects were detected between siCtrl and siTAZ alone conditions. To determine whether the increased growth mediated by TAZ-S89A in the presence of BRAFi + MEKi is dependent on TEADs, we knocked down all four TEAD paralogs in A375 TAZ-S89A cells (Fig. 4C). Under BRAFi + MEKi treatment, TEAD knockdown eliminated the

growth advantage conferred by TAZ-S89A, as there was comparable cell growth with BRAFi + MEKi treatment plus knockdown of pan-TEAD between A375 cells induced to express TAZ-S89A and non-induced cells (Fig. 4D). These results indicate that TAZ-driven tolerance to BRAFi + MEKi is dependent on TEAD expression. Furthermore, we utilized a mutant Y421E TEAD1 construct[46] that prevents YAP1/TAZ interaction but does not disrupt binding with other transcriptional co-

**Fig. 3 | TAZ dependency in melanoma. A** Scatter plot showing WWTR1 and YAP1 Chronos CRISPR gene dependency scores for melanoma cell lines (n = 62) data obtained from DepMap. Red dots indicate cell lines (n = 5) predicted to have dependence (Chronos score < -1) on WWTR1 for survival. **B** Violin plot showing WWTR1 and YAP1 Chronos CRISPR gene dependency scores for melanoma cell lines data obtained from DepMap. p < 0.0001 Two-tailed unpaired t-test. **C** Scatter plot showing WWTR1 and TEAD1 Chronos CRISPR gene dependency scores for mela-noma cell lines (n = 62) data obtained from DepMap. Pearson's correlation analysis and Welch Two Sample two-tailed t-test. **D** WM983B cells were treated with reagent alone (-), non-targeting control siRNA, siYAP1, or siTAZ for 72 hrs. Cells were lysed and lysates analyzed by Western blotting with the antibodies indicated. **E** WM983B cells were treated with reagent alone (no siRNA), non-targeting control siRNA, or siRNAs to either YAP1 or TAZ. Cells were imaged using IncuCyte Live Cell Analysis System. Shown is the mean ± SEM percent plate coverage from three independent experiments. p values from two-tailed one-way ANOVA of Area Under the Curve (AUC) analysis. ns = not significant. Source Data are available. **F** Doxycycline (DOX) inducible A375 LacZ, A375 YAP-S127A, and A375 TAZ-S89A cells were treated -/+ 100 ng/mL doxycycline for 48 hrs. Cell lysates were analyzed by Western blotting with the antibodies indicated. The experiment was repeated independently three times with similar results. **G** Immunofluorescence images of A375 HA-TAZ-S89A, cells stained for HA (green) and DAPI (blue). The experiment was performed independently three times and representative images are shown. Scale bar, 25 μm.

activators such as Vgll1, Vgll2, and Vgll3[57]. Following the induction of TEAD1-Y421E in A375 SOX10 KO cells, we detected a dramatic decrease in CYR61 and CTGF expression in both the absence and presence of BRAFi + MEKi (Fig. 4E). Furthermore, the growth of A375 4.21 SOX10 KO cells in the presence of BRAFi + MEKi was significantly reduced by induction of TEAD1-Y421E expression (Fig. 4F). TEAD1-Y421E partially reduced growth in the absence of inhibitors. These data show that TAZ-S89A is sufficient to mediate drug tolerance to BRAFi + MEKi in SOX10-positive, drug-sensitive cells in a TEAD-dependent manner.

## Synthesis and characterization of TEAD inhibitors

To improve the translational potential of our studies, we developed two TEAD inhibitors, 1-(7-(4-(trifluoromethyl)phenoxy)-3,4-dihy-droisoquinolin-2(1H)-yl)prop-2-en-1-one (OPN-9643) and 1-(7-(3-fluoro-4-(trifluoromethyl)phenoxy)-3,4-dihydroisoquinolin-2(1H)-yl)prop-2-en-1-one (OPN-9652) (Fig. 5A, B). These compounds were synthesized starting by the formation boronate ester of tert-butyl 7-bromo-3,4-dihydro-1H-isoquinoline-2-carboxylate using Suzuki Miyaura coupling followed by diphenylether formation with 4-(trifluoromethyl)phenyl boronic acid, (3-fluoro-4-(trifluoromethyl)phenyl)boronic acid and (4-fluoro-3-(trifluoromethyl)phenyl)boronic acid, respectively for OPN-9643 and OPN-9652 using Chan-Lam type coupling method. Boc removal of these intermediates, followed by acrylamide formation, provided the final compounds OPN-9643 and OPN-9652. We investi-gated these compounds in NF2-deficient mesothelioma NCI-H226 cells, which are highly sensitive to TEAD inhibition in cell proliferation and viability assays[58–60]. We treated NCI-H226 cells with increasing doses of either OPN-9643 or OPN-9652 and subsequently quantified metabolically active cells to determine growth $IC_{50}$ values. Both com-pounds impaired cell growth comparably with $IC_{50}$ values of ~100 nM (Fig. 5C, Supplementary Fig. 5A). Furthermore, both OPN-9643 or OPN-9652 potently inhibited luciferase activity in MSTO-211H cells that express a TEAD-dependent luciferase reporter[61]. OPN-9643 and OPN-9652 inhibited TEAD reporter activity with $IC_{50}$ values of 5 nM and 15 nM, respectively (Fig. 5C, Supplementary Fig. 5B). Together, growth and reporter assay data show that OPN-9643 and OPN-9652 are potent TEAD inhibitors.

To further examine OPN-9643 and OPN-9652 binding to TEAD proteins, we performed a protein thermal shift analysis and confirmed the binding of both compounds to depalmitoylated TEAD1 and TEAD4 proteins (Supplementary Fig. 5C). Incubation of these compounds with depalmitoylated TEAD4 resulted in significant Tm shifts, indi-cating binding and stabilization of target proteins. In contrast, when OPN-9643 and OPN-9652 were incubated with palmitoylated TEAD4, we did not detect binding in thermal shift assays. It has been reported that autopalmitoylation inhibitors do not alter TEAD stability[62]. Our data suggest that the compound binding site is occupied when TEAD proteins are palmitoylated. We selected TEAD1 and TEAD4 for analysis because they are the predominantly expressed TEAD paralogs in many cancer cell lines, including NCI-H226 and MSTO-211H[58] and in SOX10 KO melanoma cells (Supplementary Fig. 5D). Co-crystal structure of TEAD1 and OPN-9652 in 2.03 Å resolution revealed that the electro-philic acrylamide group of OPN-9652 reacts with Cys359 present in the

entrance pocket of TEAD1 to form a covalent bond. The natural ligand palmitic acid reacts with the same cysteine to form a thioester bond. The fluoro-trifluromethyl phenyl ether and aromatic section of the tetrahydroisoquinolin occupied the lipophilic pocket of TEAD1 where the hydrocarbon part of the palmitoyl group binds (Fig. 5D). The fluoro-trifluoromethyl group occupied the palmitoyl binding pocket in two different conformations aligning the fluoro group deep inside the pocket or on the opposite side. Together, our data suggest that OPN-9643 and OPN-9652 target the palmitoylation pocket of TEAD proteins and inhibit multiple TEAD paralogs.

## OPN-9652 and OPN-9643 reduce the expression of TEAD targets

Next, we investigated the ability of OPN-9652 and OPN-9643 to reduce the expression of TEAD-driven targets. We generated TEAD binding-driven luciferase reporter cell lines and observed that luciferase activity was significantly decreased following a 24-hour treatment with either OPN-9652 or OPN-9643 (Fig. 6A). Furthermore, Western blotting analysis detected a reduction in CTGF and CYR61 protein expression in SOX10 KO cells after OPN-9652 and OPN-9643 treatment (Fig. 6B, C and Supplementary Fig. 5E). Furthermore, OPN-9652 reduced expres-sion of CTGF and CYR61 induced by expression of TAZ-S89A in A375 cells, further underscoring the TEAD-targeting activity of OPN-9652 (Supplementary Fig. 5F).

To more broadly analyze the effects of OPN-9652 and OPN-9643, lysates from treated A375 SOX10 KO and parental cells were analyzed by Reverse Phase Protein Array (RPPA), a technique that detects alterations in signaling, cell cycle, and cell death pathways[63]. Targets that were down-regulated following OPN-9652 and OPN-9643 treat-ment included cell cycle progression regulators (FOXM1 and PLK1), growth factor receptors (phospho-cMET Y1234/5, AXL), a serine pro-tease inhibitor (PAI1), and RAD51 (Fig. 6D). Changes in proteins iden-tified from RPPA were validated via Western blot (Fig. 6E). We also tested OPN-9652 and OPN-9643 alongside a non-covalent inhibitor, VT107, and VT106, a 50x less potent enantiomer of VT107[64]. Using a luciferase TEAD reporter, we detected that OPN-9652 and OPN-9643 reduced luciferase activity comparable VT107, and significantly greater than VT106 (Supplementary Fig. 6A). To further characterize the transcriptomic effect of these inhibitors, we performed RNA-seq analysis on A375 parental and SOX10 KO cells following treatment with either OPN-9652 or OPN-9643 (Supplementary Fig. 6B). We compared RNA-seq data to five published YAP/TAZ gene signatures to further validate inhibitor efficacy. OPN-9652- and OPN-9643-treated cells showed strong overlap with previously established signatures, rein-forcing that the observed transcriptional changes are mediated by TEAD inhibition (Fig. 6F). RNA-seq analysis also detected a reduction in other known TEAD targets, such as DKK1, MYC, and TGF-β2 in SOX10 KO cells (Supplementary Fig. 6C). Our earlier findings suggest TAZ plays a more dominant role in TEAD-dependent transcription, so we compared the five most positively and negatively enriched HALLMARK gene sets following TEAD inhibitor treatment and compared those gene sets to YAP1 and TAZ knockdown (Supplementary Fig. 6D). These data show there is a greater overlap between TEAD inhibition and TAZ depletion, whereas there is little enrichment shared with YAP1

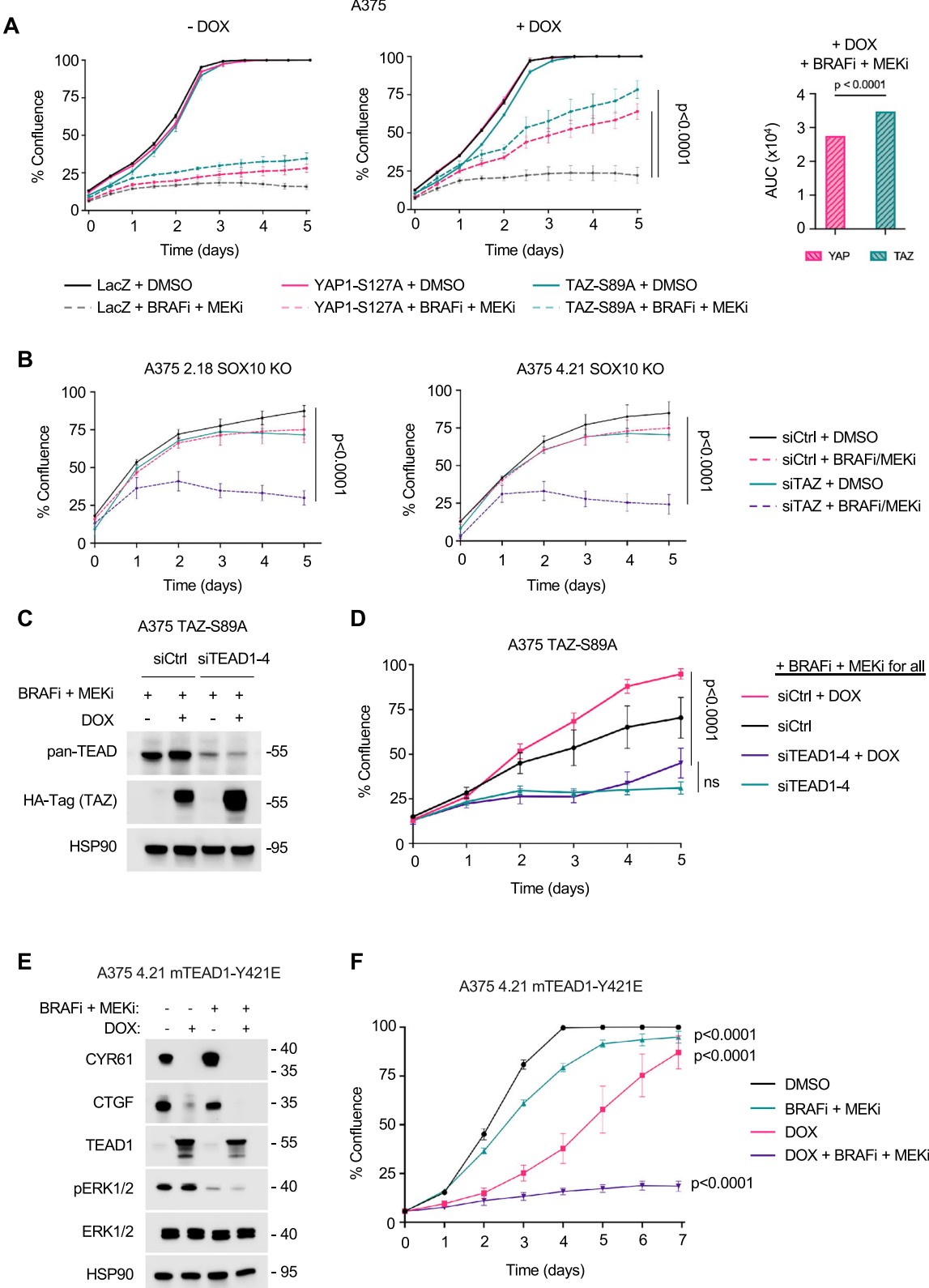

depletion. These results show that OPN-9652 and OPN-9643 reduce TEAD target expression in SOX10 KO melanoma cells.

## OPN-9652 and OPN-9643 enhance the effects of MAPK-targeted therapy in SOX10 KO cells

Despite the changes in cell cycle-related proteins, treatment with OPN-9652 or OPN-9643 alone did not significantly decrease cell growth in SOX10 KO cells by IncuCyte assay (Supplementary Fig. 7A). Since A375 SOX10 KO cells are tolerant to BRAFi + MEKi[18], and TAZ and YAP1 induce drug-tolerance in A375 parental cells, we tested whether OPN-9652 and OPN-9643 sensitized cells to combination BRAFi + MEKi. We treated A375 SOX10 KO clones with a triple combination of TEADi and BRAFi + MEKi in a 2D IncuCyte assay to measure cell growth. When administered in combination with BRAFi + MEKi, OPN-9652 or OPN-

**Fig. 4 | TAZ is required for TEAD signaling. A** A375 LacZ, A375 YAP1-S127A, and A375 TAZ-S89A cells were induced with 100 ng/mL of doxycycline (DOX) and treated with either 1 μM PLX4270, 35 nM PD-0325901 or vehicle control. Cells were imaged using IncuCyte Live Cell Analysis System. Treatment was renewed every 48-72 hrs. Shown is the mean ± SEM from three independent experiments. Statistics are one-way ANOVA of AUC analysis. Source Data are available. **B** 2.18 SOX10 KO and 4.21 SOX10 KO cells were treated with a non-targeting control siRNA, TAZ siRNA, 1 μM PLX4270 and 35 nM PD-0325901, or both TAZ siRNA and 1 μM PLX4270 and 35 nM PD-0325901. Cells were imaged using IncuCyte Live Cell Analysis System. Shown is the mean ± SEM percent plate coverage from three independent experiments. p values from one-way ANOVA of AUC analysis. Source Data are available. **C** A375 TAZ-S89A cells were induced with -/+ 100 ng/mL doxycycline and treated with -/+ 1 μM PLX4270 and 35 nM PD0325901, or vehicle control for 72 hrs following the knockdown of TEADs 1-4. Cell lysates were analyzed by Western blotting with the

antibodies indicated. **D** A375 TAZ-S89A cells were induced with 100 ng/mL of doxycycline and subsequently treated with 1 μM PLX4270 and 35 nM PD0325901, or vehicle control -/+ siRNAs for TEADS 1-4. Cells were imaged using IncuCyte Live Cell Analysis System. Treatment was renewed every 48 hrs. Shown is the mean ± SEM from three independent experiments. ****p < 0.0001 one-way ANOVA of AUC analysis. ns = not significant. Source Data are available. **E** A375 4.21 mTEAD-Y421E cells were induced with -/+ 100 ng/mL doxycycline and treated with -/+ 1 μM PLX4270 and 35 nM PD0325901, or vehicle control for 24 hrs. Cell lysates were analyzed by Western blotting with the antibodies indicated. **F** A375 4.21 mTEAD-Y421E cells were induced with 100 ng/mL of doxycycline and subsequently treated with 1 μM PLX4270 and 35 nM PD0325901, or vehicle control. Cells were imaged using IncuCyte Live Cell Analysis System. Treatment was renewed every 48 hrs. Shown is the mean ± SEM from three independent experiments. ****p < 0.0001 one-way ANOVA of AUC analysis. Source Data are available.

9643 markedly decreased cell growth in A375 SOX10 KO cells compared to BRAFi + MEKi alone (Fig. 7A, Supplementary Fig. 7B). In MEKi tolerant MeWo SOX10 KO cells, MEKi + OPN-9652/OPN-9643 combination also significantly reduced cell growth (Fig. 7B). Analysis of IncuCyte live-cell imaging masks further confirmed the dramatic reduction in cells (Supplementary Fig. 7C). To further extend our analysis, we probed for markers of cell death pathways in MeWo SOX10 KO cells and detected an increase in cleaved-PARP and cleaved-GSDME in the combination treated cells (Fig. 7C).

Next, we used 3D spheroids that better mimic in vivo tumors through their enhanced cell-cell interactions, tissue architecture, hypoxic core, and nutrient gradients compared to a 2D monolayer[65–67]. Compared to vehicle controls, OPN-9652 alone did not alter live (calcein AM-green) and dead (PI-red) staining; however, in combination with BRAFi + MEKi, OPN-9652 significantly reduced outgrowth area and increased cell death compared to DMSO (p < 0.05) and TEADi alone (p < 0.05) conditions in both 2.18 and 4.21 cells (Supplementary Fig. 8A–C). Although not statistically significant, the outgrowth area was decreased, and cell death was increased in SOX10 KO cell spheroids treated with triple combination therapy versus BRAFi + MEKi treatment. Finally, we tested the ability of TEADi to reverse the effects of active TAZ-mediated tolerance to BRAFi + MEKi. Expression of TAZ-S89A, enhanced growth of A375 cells in the presence of BRAFi + MEKi, which reversed albeit incompletely by OPN-9652 (Fig. 7D). These data show that TEADi sensitizes SOX10 KO drug-tolerant cells to BRAFi + MEKi treatment.

## Targeting TEAD delays the onset of acquired resistance to BRAFi + MEKi

Based on the enhanced effect with BRAFi + MEKi, we tested OPN-9652 and OPN-9643 in acquired-resistance models. We have previously generated in vivo acquired-resistance tumor cell lines to the combination of BRAFi + MEKi (combination resistant tumors denoted as CRTs) derived from A375 cells[68] and to the paradox breaker BRAFi, PLX8394 (Paradox Breaker Resistant Tumor denoted as PBRT) derived from 1205Lu cells[69]. GSEA of RNA-seq showed a positive enrichment of the mesenchymal-like state in CRT34 and CRT35 cells compared to A375 parental cells, and in PBRT15 and PBRT16 cells compared to 1205Lu parental cells (Fig. 8A). CRT and PBRT cell lines lose SOX10 expression during the acquisition of resistance[18] and exhibited enrichment in YAP1/TAZ gene signatures compared to their SOX10-expressing parental counterparts (Fig. 8B). Furthermore, the TEAD targets, CYR61 and CTGF, were up-regulated in the CRT34 and CRT35 cell lines compared to parental cells (Fig. 8C). These data further emphasize the relationship between SOX10 loss, YAP1/TAZ-TEAD signaling, and insensitivity to MAPK targeted therapy.

To test whether TAZ is the primary mediator of pathway activity in CRT cell lines, we selectively knocked down either TAZ or YAP1. In both CRT34 and CRT35, we consistently detected a greater reduction in CTGF and CYR61 expression when knocking down TAZ compared to

YAP1 (Fig. 8D). Thus, similar to SOX10 KO cells, TEAD activity in acquired resistant cell lines primarily depends on TAZ. We detected decreased levels of CYR61 and CTGF via Western blot analysis after OPN-9652 and OPN-9643 treatment in CRT34 and CRT35 cells (Fig. 8E). To validate whether OPN-9652 reduced the expression of TEAD targets in vivo, we treated A375 4.21 SOX10 KO-derived xenografts for 3 days, and evaluated CYR61 expression levels by Western Blot. CYR61 levels were significantly reduced in SOX10 KO xenografts following OPN-9652 treatment (Fig. 8F, G).

Drug tolerance emerges as a reservoir of cells during MRD. Previously, our lab has shown that BRAF mutant xenograft models progress to MRD within 4 weeks of BRAFi + MEKi treatment[70]. To replicate this model, we utilized A375 parental xenografts treated with BRAFi + MEKi for 4 weeks until tumor size did not dramatically change for 2 weeks, which we considered MRD. Then, mice received either BRAFi + MEKi or the triple combination of BRAFi, MEKi plus OPN-9652. Tumors on BRAFi + MEKi progressed to an average of 750 mm³ by day 54 (Fig. 8H, Supplementary Fig. 9A). By contrast, tumors in the triple combination arm regrew very slowly, and only one tumor reached the 750 mm³ threshold by day 71. Mouse survival was significantly improved by adding OPN-9652 to BRAFi + MEKi in A375 xenografts (Fig. 8I). We also tested the combinations in a quickly regressing model, mouse BRAF V600E mutant YUMM1.7 tumors in NSG mice. While many mouse melanoma cell lines express Sox10, YUMM1.7 cells have low Sox10 expression[71] and exhibit high Cyr61 expression (Supplementary Fig. 9B). YUMM1.7 tumors were treated with BRAFi + MEKi chow until tumors regressed. From the nadir of tumor size, a triple combination of BRAFi, MEKi, and OPN-9652 extended median survival compared to BRAFi + MEKi alone from 21.5 days to 27 days (Supplementary Fig. 9C, D). These data show that TEAD activity is maintained in acquired resistant cells and that TEADi may be used to delay the onset of acquired resistance in vivo.

## Discussion

Drug-tolerant persister cells underlie MRD[1,72]. Identifying and targeting vulnerabilities within persister cells before acquired resistance is likely to improve the efficacy of targeted therapies. Several resistance mechanisms to MAPKi have been identified, including enhanced PI3K/AKT signaling[73], expression of BRAF splice variants[74], and NF1 loss[75,76]. Our lab previously showed that the loss of SOX10 is sufficient to induce a drug-tolerant phenotype and alters pathways such as TGFβ signaling and TNFα signaling via NFκB[18]. Here, we identified that up-regulation of TEAD following the loss of SOX10 is highly dependent on TAZ. We define differences between TAZ-dependent and YAP1-dependent transcriptomes and show that exogenous TAZ-S89A expression is sufficient to induce tolerance to MAPKi in drug-naïve SOX10-expressing cells. We generated two compounds, OPN-9652 and OPN-9643, that inhibit TEADs, resensitize SOX10 KO cells to BRAFi + MEKi, and delay the onset of tumor resistance to BRAFi + MEKi from MRD. Our research highlights the significance of recognizing vulnerabilities in

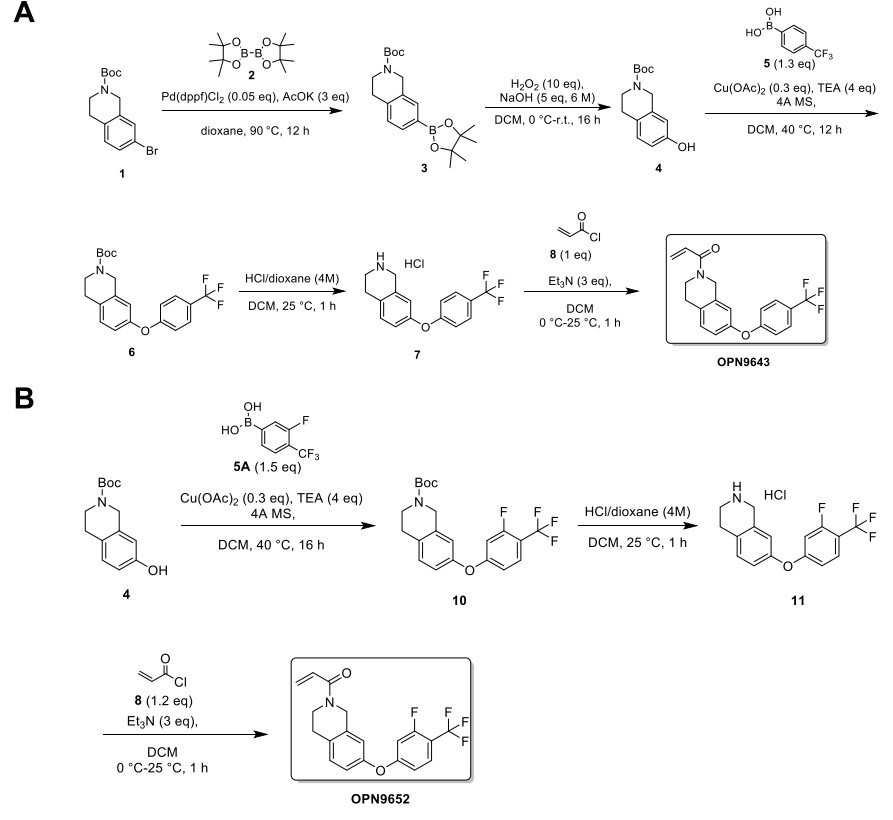

**A**

**B**

**C**

| Compound | MSTO-211H TEAD Reporter IC$_{50}$ (µM) | NCI-H226 Growth IC$_{50}$ (µM) | Delta Tm TEAD1/TEAD4 (°C) |
|---|---|---|---|
| OPN-9652 | 0.005 | 0.093 | 12.8/7.8 |
| OPN-9643 | 0.015 | 0.123 | 13.5/7.0 |

**D**

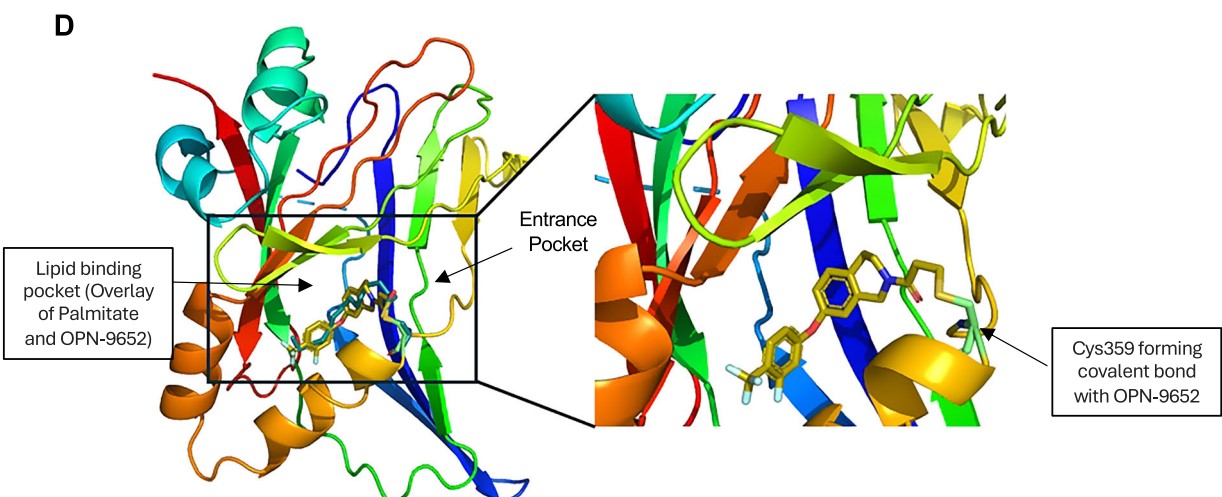

**Fig. 5 | Chemical synthesis and characterization of TEAD inhibitors. A** The steps in the chemical synthesis of OPN-9643. **B** The steps in the synthesis of OPN-9652. **C** Results of MSTO-211H reporter assay IC$_{50}$ values, NCI-H226 growth IC$_{50}$ values, and Delta Tm TEAD1/TEAD4 protein thermal shift values following OPN-9643 and OPN-9652 treatment. **D** Co-crystal structure of OPN-9652 bound to TEAD1 (PDB ID: 8S6Y). OPN-9652 occupies the central palmitate-binding pocket and covalently modifies Cys359. Image on the left shows an overlay of OPN-9652 (gold) and palmitate (green). Image on the right shows a zoomed-in view covalent modification of Cys359 (mint) by OPN-9652 (gold).

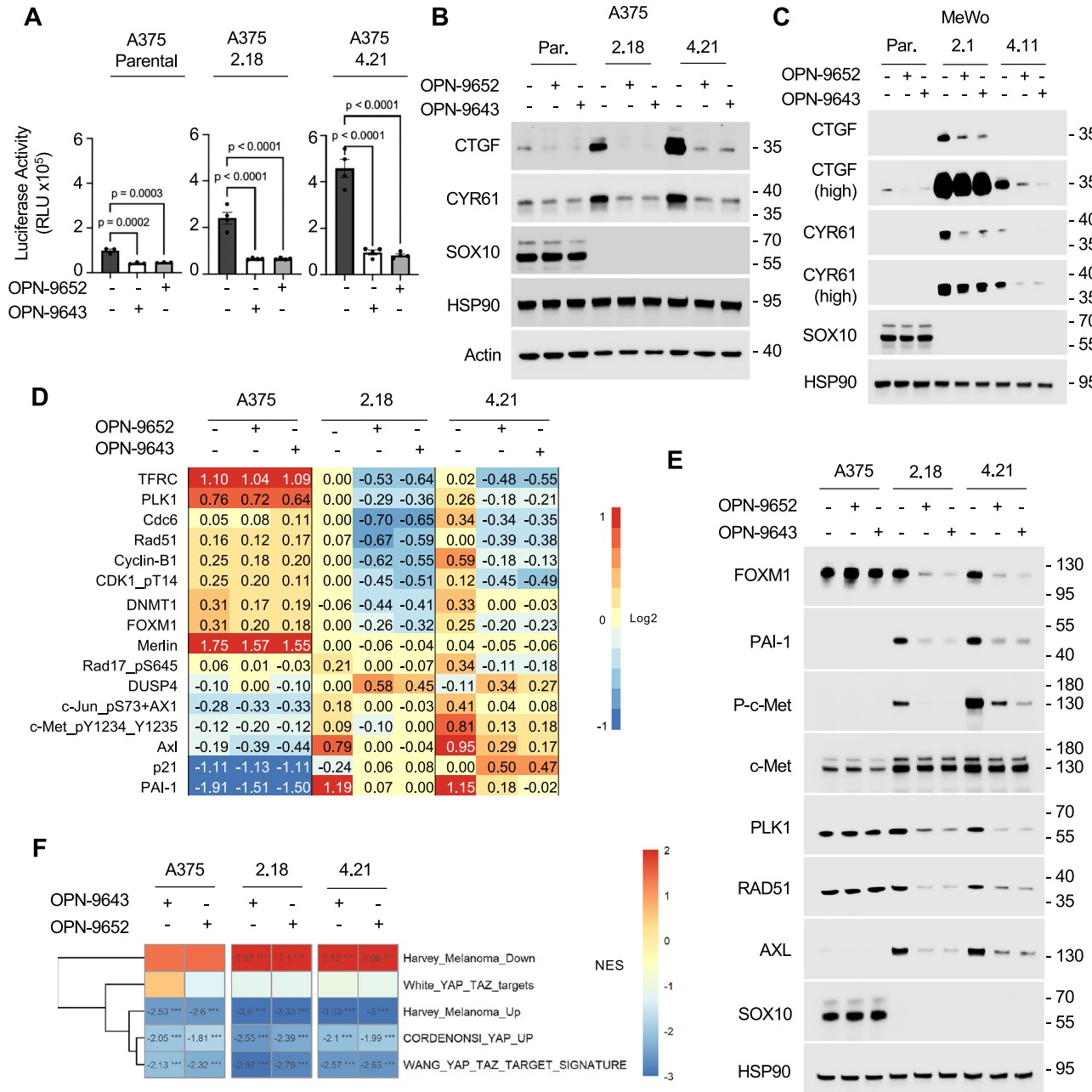

**Fig. 6 | OPN-9652 and OPN-9643 inhibit TEAD-driven transcriptional activity.**
**A** A375 parental, A375 crSOX10 #2.18, and A375 crSOX10 #4.21 cells were treated
with 2 μM of either OPN-9652 or OPN-9643 for 24 hrs and then cells were lysed.
Firefly luciferase activity was measured via Dual-Luciferase® Reporter Assay Sys-
tem. The experiment was repeated independently as biological replicates three
times with similar results. Data are expressed as mean ± SEM. ***p < 0.001,
****p < 0.0001, One-way ANOVA. **B** A375 parental, A375 crSOX10 #2.18, and A375
crSOX10 #4.21 cells were treated with 2 μM of either OPN-9652 or OPN-9643 for
24 hrs. Cell lysates were analyzed by Western blotting with the antibodies indicated.
The experiment was repeated independently three times with similar results.
**C** MeWo parental, MeWo crSOX10 #2.1, and MeWo crSOX10 #4.11 cells were treated
with 2 μM of either OPN-9652 or OPN-9643 for 24 hrs. Cell lysates were analyzed by
Western blotting with the antibodies indicated. The experiment was repeated

independently three times with similar results. **D** A375 crSOX10 #2.18, and A375
crSOX10 #4.21 cells treated with 2 μM of either OPN-9652 or OPN-9643 for 48 hrs,
lysed, and processed for RPPA. Shown is a heat map from three independent
experiments showing median-centered log2-transformed group average expres-
sion data for antibodies with an absolute log-2 fold change >1 following treatment.
**E** Lysates from A375 crSOX10 #2.18, and A375 crSOX10 #4.21 cells treated with 2 μM
of either OPN-9652 or OPN-9643 for 48 hrs were analyzed by Western blotting with
the antibodies indicated. The experiment was repeated independently three times
with similar results. **F** Heatmap showing NES of YAP1/TAZ gene signatures following
treatment of either OPN-9652 or OPN-9643 in A375 parental and SOX10 KO cell
lines compared to vehicle-treated cells. *p < 0.05, **p < 0.01, ***p < 0.001 two-
tailed BHFDR.

drug-tolerant persister cells and provides a strategy to target MRD that
enhances the durability of current standard care therapies for
melanomas.

Melanoma exhibits multiple phenotypic states with distinct tran-
scriptional programs independent of genotype[12,45,77]. The invasive

phenotype is marked by tolerance to targeted therapy, low levels of
SOX10 expression, and high AP-1 and TEAD activities[45,77]. We showed
that SOX10 loss is associated with reduced Merlin expression, a key
regulator of the Hippo signaling pathway, but that additional
mechanisms occur. How SOX10 regulates Merlin and additional

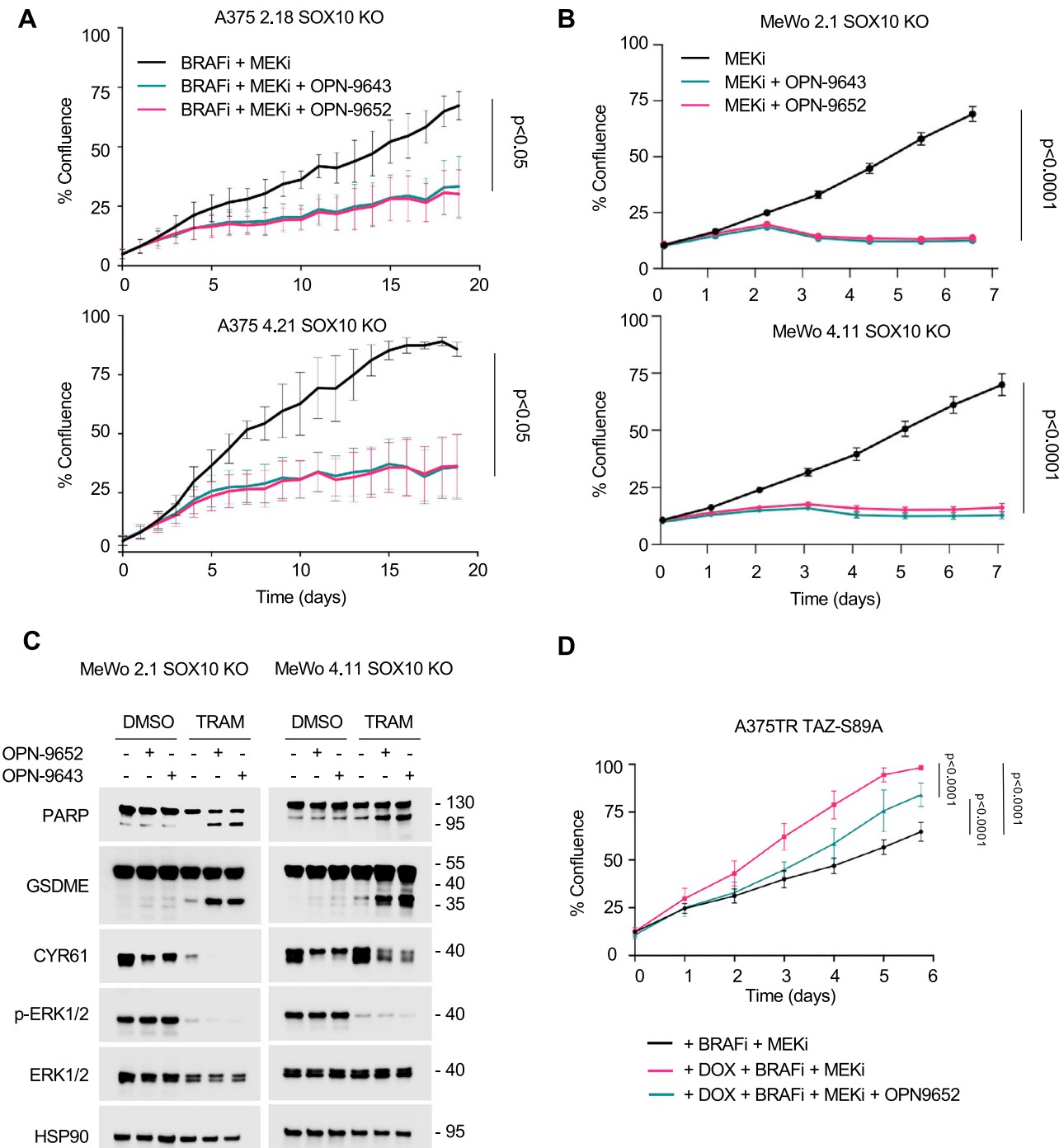

**Fig. 7 | OPN-9652 and OPN-9643 enhance MAPK-targeted therapy. A** A375 crSOX10 #2.18 and A375 crSOX10 #4.21 cells were treated with 1 μM PLX4270, 35 nM PD0325901, and 2 μM of either OPN-9652 or OPN-9643 and imaged using the IncuCyte Live Cell analysis system. Treatment was renewed every 48-72 hrs. Cell growth was determined as percent plate coverage. Shown is the mean ± SEM from three independent experiments. Source Data are available. **B** MeWo crSOX10 #2.1 and MeWo crSOX10 #4.11 cells were treated with 50 nM of Trametinib, and 2 μM of either OPN-9652 or OPN-9643 and imaged using the IncuCyte Live Cell analysis system. Treatment was renewed every 48-72 hrs. Cell growth was determined as percent plate coverage. Shown is the mean ± SEM from three independent experiments. Source Data are available. **C** Lysates from MeWo crSOX10 #2.1 and MeWo crSOX10 #4.11 cells treated with 50 nM of Trametinib, 2 μM of either OPN-9652 or OPN-9643, or vehicle control for 48 hrs were analyzed by Western blotting with the antibodies indicated. The experiment was repeated independently three times with similar results. **D** A375TR TAZ-S89A cells were induced with 100 ng/mL of doxycycline and treated with 1 μM PLX4270, 35 nM PD0325901 (BRAFi + MEKi) and either OPNA-9652 or vehicle control. Cells were imaged using IncuCyte Live Cell Analysis System. Treatments were renewed every 48-72 hrs. Shown is the mean ± SEM from three independent experiments. Statistics are one-way ANOVA of AUC analysis. Source Data are available.

mechanisms of SOX10 regulation of TEAD activity, including the increased expression of c-Jun and increased chromatin accessibility following SOX10 loss, warrant further analysis. The TEAD co-activators, TAZ and YAP1, are highly conserved with ~50% homology at the amino acid level[78]. Despite this homology, there are structural differences between TAZ and YAP1. TAZ lacks a hydrophobic residue within the helix-loop-helix TEAD binding domain that is present in YAP1[79]. TAZ has been shown via crystal structure to form a tetrameric complex with

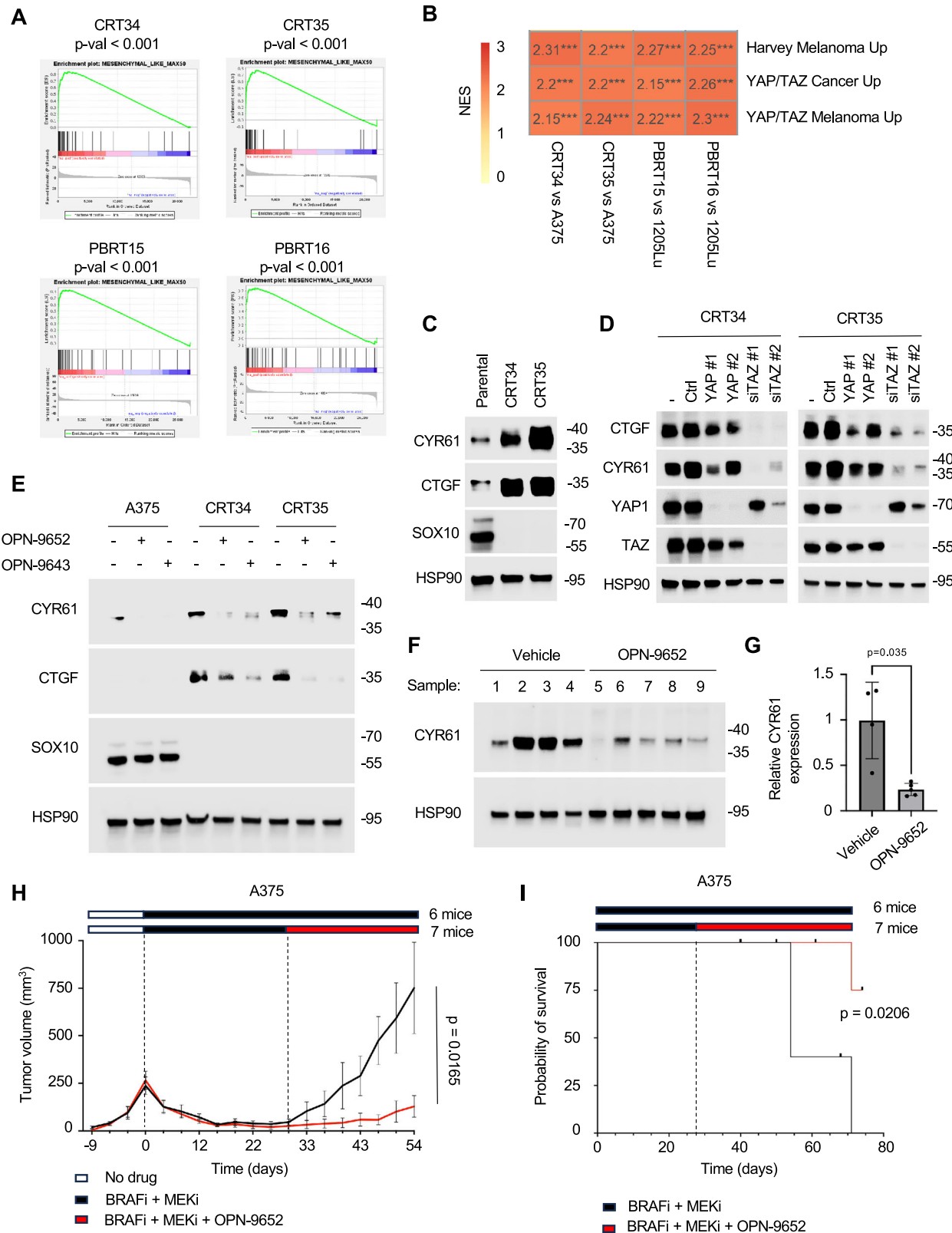

TEAD4 in a 2:2 ratio, a complex not observed with YAP1[80]. Despite these structural differences, the different roles of YAP1 and TAZ play remain poorly defined. We show that TEAD targets are highly dependent on TAZ in SOX10-deficient melanoma. Depletion of TAZ reduces more genes than YAP1 depletion, and the reduced genes are enriched for TEAD target genes. One possibility is that TAZ may be able to compensate for the loss of YAP1 but not vice versa in our model. Other studies show that YAP1, but not TAZ, mediates TEAD target expression and resistance to CDK4/6 inhibitor in breast cancer cells[81] and that TAZ determines PD-L1 expression in breast and lung cancer[82]. Overexpression of active YAP1 is sufficient to promote resistance to BRAFi in cancer[32,83]. Here, we show that TAZ is sufficient to induce tolerance to BRAFi + MEKi in melanoma, and that targeting TAZ resensitizes drug-tolerant cells to targeted therapy.

**Fig. 8 | YAP1/TAZ-TEAD signaling up-regulation is TAZ-dependent in acquired resistant melanoma cell lines. A** Enrichment plots of mesenchymal signature[38] for CRT34 and CRT35, and PBRT15 and PBRT16, cells compared vs parental cells. ***p < 0.001 two-tailed BHFDR. **B** Heatmap showing GSEA of YAP1/TAZ gene signatures comparing parental A375 and 1205Lu cell lines to drug-tolerant cell lines. *p < 0.05, **p < 0.01, ***p < 0.001 two-tailed BHFDR. **C** A375 parental, CRT34, CRT35, A375 2.18 SOX10 KO, and A375 4.21 SOX10 KO cells were plated and left untreated. Cell lysates were analyzed by Western blotting with the antibodies indicated. The experiment was repeated independently three times with similar results. **D** CRT34 or CRT35 cells were treated with reagent alone, non-targeting control siRNA, siYAP1, or siTAZ for 72 hrs. Lysates were analyzed by Western blotting with the antibodies indicated. The experiment was repeated independently three times with similar results. **E** A375 parental, CRT34, and CRT35 cells were treated with 2 μM of either OPN-9652 or OPN-9643 for 48 hrs. Cells were lysed, and lysates were analyzed by Western blotting with the antibodies indicated. The experiment was repeated independently three times with similar results. **F** 4.21 SOX10 KO cells were injected into NSG mice. Once tumors reached 100 mm³, mice received either OPN-9652 (50 mg/kg) for 3 days or vehicle. Tumors were harvested and analyzed by Western blotting with the antibodies indicated. **G** Quantification of CYR61 expression performed in ImageLab using volume tools to calculate CYR61 compared to HSP90 loading control. Data are expressed as mean ± SD Vehicle n = 4, OPN-6952 n = 5. *p < 0.05, Welch's two-tailed unpaired t-test. **H** Mean tumor growth graph comparing treatment arms of A375 xenograft treated with PLX4720 (200 PPM), PD0325901 (7 PPM) alone (6 mice) or in combination with OPN-9652 (50 mg/kg) (7 mice). Data are expressed as mean ± SEM. Welch's two-tailed t-test. Source Data are available. **I** Mouse survival curves from experiment in Fig. 8H, indicating time for tumor to reach 1000 mm³. Censored mice were due to unexplained deaths. P value was calculated using Logrank (Mantel-Cox) test. Source Data are available.

Previous studies have utilized verteporfin to inhibit YAP1/TAZ-TEAD to overcome BRAFi resistance in melanoma[84]; however, verteporfin is a non-selective pathway inhibitor. All TEAD family proteins undergo autopalmitoylation, which is necessary for protein stability[85], and the autopalmitoylation binding pocket has been targeted by small molecule inhibitors[64]. Here, we develop two moieties, OPN-9652 and OPN-9643, that covalently bind to the palmitate-binding pocket of TEAD 1-4 and prevent autopalmitoylation. Both OPN-9652 and OPN-9643 effectively reduced TEAD-driven luciferase activity and canonical TEAD targets, CTGF and CYR61, indicating their on-target effects. Furthermore, they resensitized drug-tolerant SOX10 KO cells to BRAFi + MAPKi in vitro, and prolonged the durable response in vivo likely by targeting drug-tolerant persister cells.

Inhibition of TEADs holds promise in the clinic. There are multiple TEADi currently in Phase I human clinical trials for NF2-deficient mesotheliomas that target the highly-conserved central palmitate binding pocket, including VT3989 from Vivace Therapeutics (NCT04665206)[86]. Another TEADi, IAG933 from Novartis Pharmaceuticals, which directly disrupts protein-protein interactions between either YAP1/TAZ and TEAD, is also being tested (NCT04857372). Three interfaces mediate the binding of YAP1/TAZ to TEAD, and IAG933 targets interface 3[87]. While TEADi is currently mainly in clinical trials in mesothelioma with Hippo pathway alterations, our studies form the basis for the potential use of TEADi in melanoma.

Our data also implicate TAZ as the major co-activator of TEADs in cutaneous melanoma, but the inhibitors developed hold therapeutic potential for targeting YAP1 and/or TAZ-TEAD signaling in other cancers. YAP1 is implicated as a driver of resistance in KRAS-mutant cancers. In KRAS-mutant pancreatic cancer, YAP1 expression correlates with worse relapse-free survival, and verteporfin enhanced the effects of a RAF inhibitor[88]. YAP1 rescued cell survival in KRAS-dependent colon cancer following KRAS-ablation[89], and in KRAS-mutant lung cancer, YAP1 expression promoted resistance to MEK inhibitors[90]. A recent study showed that the YAP/TAZ-TEAD transcriptome is up-regulated following acquired resistance to Sotorasib in KRAS[G12C] NSCLC cell lines, and combination therapy with a pan-TEAD inhibitor, GNE-7883, resensitizes the cells to targeted therapy[91]. Thus, targeting TEADs holds promise to broadly enhance the effects of targeted therapies in many cancers.

Together, our findings support a model in which SOX10 loss promotes a shift toward a drug-tolerant melanoma cell state characterized by increased reliance on TEAD-driven transcription and sensitivity to TEAD inhibition, indicating a context-dependent vulnerability. Our findings highlight a potential clinical strategy in which standard-of-care therapy is used initially to reduce tumor burden, followed by addition of a TEAD inhibitor at the minimal residual disease stage, which is designed to eliminate drug-tolerant persister cells.

## Methods

### Cell culture
A375 parental (purchased from ATCC in 2005), A375-YAP1 S127A, A375-TAZ S89A, or A375-LacZ, MeWo parental (donated by Dr. Barbara Bedogni, when at Case Western Reserve, Cleveland, OH in 2014), and previously generated crSOX10 KO cells[18] were cultured in DMEM with 10% FBS and 1% penicillin/streptomycin. A375 BRAFi + MEKi tolerant cells (CRT34, CRT35, previously referred to as CRT14 and CRT15)[68] were cultured in the presence of PLX4720 (1 μM) and PD-0325901 (35 nM). WM983B cells were cultured in MCDB153 (Sigma) with 2% FBS, 20% Leibowitz L-15 medium, and 5 μg/ml insulin. *BRAF* and *NRAS* mutation status in cell lines were validated by Sanger sequencing. Cells were tested monthly for mycoplasma contamination with the MycoScope Kit (Genlantis). Short-tandem repeat analysis was completed for MeWo parental, MeWo crSOX10 KO cells A375 parental, A375 crSOX10 KO cells, and CRT cells. All cell lines matched a known profile. Cells were cultured at 37 °C with 5% CO₂ in a humidified chamber.

### Western blot analysis
Proteins were extracted with Laemmli sample buffer, resolved by SDS-PAGE, and transferred to PVDF membranes. Immunoreactivity was detected using HRP-conjugated secondary antibodies (CalBioTech, Spring Valley, CA) and chemiluminescence HRP-recognizing substrates (ThermoScientific, Waltham, MA) on a VersaDoc Multi-Imager. Primary antibodies SOX10 (#89356, 1:1000) CTGF (#86641, 1:1000), CYR61 (#14479S, 1:1000), HSP90 (#4877, 1:3000), Merlin (#12888, 1:1000), TAZ (#70148S, 1:1000), HA-Tag (#2367, 1:1000), TEAD1 (#12292S, 1:1000), P-ERK1/2 (Thr202/Tyr204) (#9101, 1:1000), Total ERK1/2 (#9102, 1:1000) pan-TEAD (#13295S, 1:1000), FOXM1 (#5436, 1:1000), PAI-1 (#49536, 1:1000), and P-c-Met (Tyr1234/1235) (#3077, 1:1000), and c-Met (#8198, 1:1000), PLK1 (#4513S, 1:1000), RAD51 (#8875, 1:1000), and AXL (#8661, 1:1000) were purchased from Cell Signaling Technology. β-Gal (#Z378A, 1:1000) was purchased from Promega. YAP1 (#ab52771, 1:1000), TEAD4 (#ab58310, 1:1000) antibodies were purchased from Abcam. Actin (#A2066, 1:2000) antibodies were purchased from Santa Cruz Biotechnology. Secondary antibodies Goat Anti-Mouse IgG (#401215, 1:4000) and Goat Anti-Rabbit IgG (#401315, 1:4000) were purchased from Sigma-Aldrich Co.

### Small interfering RNA (siRNA) transfection
A375 crSOX10 KO cells, MeWo crSOX10 KO cells, and CRT cells were transfected with siRNA targeting YAP1 (Dharmacon, #D-012200-01, #D-012200-02), WWTR1 (Dharmacon, #D-016083-01, #D-016083-02), TEAD1 (J-012603-05), or TEAD4 (J-019570-08) at a final concentration of 25 nM using Lipofectamine RNAiMAX (Invitrogen). The following siRNA sequences were used: WWTR1 #1 GACAUGAGAUCCAUCACUA; WWTR1 #2 GGACAAACACCCAUGAACA; YAP1 #1 GGUCAGAGAUACUUCUUAA; YAP1 #2 CCACCAAGCUAGAUAAAGA. TEAD1 #5 CGAUUUGUAUACCGAAUAA; TEAD4 #8 GACAGAGUAUGCUCGCUAU.

## Lentiviral constructs

To construct doxycycline-inducible YAP1 S127A, TAZ S89A, and LacZ cDNA lentiviruses, pLentipuro/TO/V5-DEST was used. HA-tagged TAZ S89A, and control LacZ were cloned into pLentipuro/TO/V5-DEST using NEBuilder HiFi assembly cloning kit (E5520S). The mouse TEAD1 Y421E sequence was amplified via PCR and the fragment was then inserted into a pVADE2 backbone using the NEBuilder HiFi assembly cloning kit. Expression constructs and packaging plasmids pLP1, pLP2, and pLP/VSVG were co-transfected into HEK293FT cells to generate viral particles. Cells were transduced for 72 hrs, then selected with puromycin. Transgene expression was induced with doxycycline (0.1 μg/ml).

## Cell growth assay

Cells were plated in the wells of six-well plates. Sixteen pictures per well were taken every 2 hrs and data are a representation of percent plate coverage. To analyze the effects of OPN-9652 and OPN-9643 on the growth of NCI-H226, cells were plated at low density in growth medium (RPMI-1640, 10% FBS, 1% Pen/Strep) in 96-well plates. Cells were incubated in medium containing OPN-9652 and OPN-9643 for 5 days before CellTiter Glo viability measurements (Promega). These experiments were performed in duplicate.

## DepMap data

Chronos CRISPR gene dependency score[53] and gene expression[92] data were obtained from the Cancer Dependency Map (DepMap) v22Q2 (https://depmap.org/portal/). Melanoma cell lines were called dependent on a gene if the Chronos score was below -1, as recommended by DepMap. The multimode package (https://cran.r-project.org/package=multimode) was used to test for uni-modality in SOX10 gene expression and identify the anti-modal position for bimodal data. Chronos scores for YAP1, TAZ and TEAD1 were compared between SOX10 high (n = 53) and low (n = 8) groups. Pearson's correlation analysis and Welch Two Sample t-test was performed using the stats package (v4.3.2) (https://www.R-project.org). Analyses were performed using R (v4.3.2 https://www.R-project.org) and Rstudio (v2023.6.1 https://www.posit.co).

## Inhibitors

PD-0325901 for in vitro experiments was purchased from Selleck Chemicals. VT106 and VT107 in the Supplementary Data were a kind gift from Dr. Tracy Tang (Vivace Therapeutics).

## Bulk RNA-seq sample prep, data acquisition, and analysis

A375 parental, crSOX10#2 and crSOX10#4 knockout cells that were untreated, transfected with siCtrl, siYAP1#1, siYAP1#2, siTAZ#1, or siTAZ#2 were sent in triplicates for RNA sequencing. A 200 ng aliquot of each sample was transferred into library preparation, which uses an automated variant of the Illumina TruSeq™ Stranded mRNA Sample Preparation Kit. The final libraries were sequenced on Illumina NovaSeq 6000 using 101 bp paired-end with an eight-base index barcode read. Raw FASTQ RNA sequencing reads were aligned to the GRCh38 human reference genome using Star method[93] and GENCODE[94] annotations. RSEM[95] was used to quantify gene-level expression. The RNA-seq data generated for this publication can be found under GEO accession numbers GSE259388 (YAP1/TAZ knockdown in melanoma) and GSE259389 (TEADi treatment in melanoma cells).

## Bulk RNA-seq datasets

MeWo and A375 parental and CRISPR SOX10 knockout (guide #2 and #4) samples, as well as BRAFi + MEKi-resistant cell lines (CRT#34 and CRT#35) from Sanchez et al.[68]. and 1205 LuTR parental and PBRT resistant samples (PBRT#15 and PBRT#16) from Hartsough et al.[69] were previously processed from raw sequencing reads to generate pre-ranked lists with DESeq2 test statistic values in Capparelli et al.[18]. RNA-seq data for three parental and SOX10 knockdown samples originating

from Sun et al.[13]. were previously processed from raw sequencing reads to generate a pre-ranked list with log2-transformed ratio values[96].

## Gene set enrichment analysis

The Gene Set Enrichment Analysis (GSEA) method[97,98] was used to identify significantly altered pathways. GSEA was implemented using the GSEA software (https://www.gsea-msigdb.org) or fGSEA package in R (fGSEA; 1.24.0 https://bioconductor.org/packages/release/bioc/html/fgsea.html). The MSigDB Hallmark gene set collection (v7.5.1)[99] and Yap/Taz gene signatures from Harvey et al.[47], White et al.[51], and Cordenonsi et al.[50]. were used for analyses. Unless noted otherwise, the DESeq2 Wald test statistic was used as a ranking metric to perform GSEA in pre-ranked mode, with genes having zero base mean or "NA" test statistic values filtered out to avoid providing numerous duplicate values. GSEA-preranked analysis was performed using the "weighted" enrichment statistic. The number of permutations was set to 1000 and FDR q-values equaling zero are reported as less than 0.001. The minimum and maximum gene set sizes were set to 15 and 500, respectively. Heatmaps were generated using the pheatmap (v1.0.12 https://cran.r-project.org/web/packages/pheatmap/index.html) package.

## ATAC-seq analysis

Raw ATAC sequencing data were downloaded from the SRA using the SRA toolkit (v2.11.1)[100,101] for melanoma cell lines within SRA accession SRP215051[39]. The Nextflow[102] core atacseq pipeline (v 2.1.2) (https://github.com/nf-core/atacseq) was used for QC, aligning reads using bwa[103], MACS2 peak-calling[104], and generating consensus peak counts using featureCounts[105]. Reads were aligned to the GRCh37 reference genome, and ENSEMBL regulatory database (v114)[106] was used for identifying promoter and enhancer regions. Consensus peaks with at least 10 counts were retained for further analysis. Differential peak expression analysis between SOX10-negative and SOX10-positive cell lines was performed using the DESeq2 package (v1.40.2)[107]. JASPAR 2020[108] motifs were used when performing Binary Motif Enrichment Analysis via monaLisa (v1.6.0)[109] for peaks appearing in promoter or enhancer regions. Dot plots were generated using ggplot2 (v3.5.1 https://cran.r-project.org/package=ggplot2). R (v 4.3.2 https://www.r-project.org/) code was implemented using RStudio (v2023.6 https://www.posit.co/).

**YAP1 and TAZ signatures:** Differential expression analysis was performed between siYAP1#1, siYAP1#2, siTAZ#1 or siTAZ#2 and siControl for A375 crSOX10#2 and crSOX10#4 cell lines using the Wald test in DESeq2[107]. Genes were considered differentially expressed (DE) if they had BHFDR ≤ 0.05 and an absolute log2 fold change ≥ 1. YAP1 and TAZ DE gene lists were initially filtered to those commonly altered in crSOX10 cell lines as well as their respective siRNAs. Lists were further refined by filtering genes found to be in common with YAP1 and TAZ CHIP-seq data[52] to create four YAP1/TAZ signatures. Venn diagrams, heatmaps and scatter plots were generated using the ggplots (v3.1.3 https://cran.r-project.org/web/packages/gplots/index.html), heatmap (v1.0.12 https://cran.r-project.org/web/packages/pheatmap/index.html) and ggplot2 (v3.4.2 https://ggplot2.tidyverse.org/) packages, respectively. Fisher's exact test was used to determine the enrichment of publicly-available Yap and Taz gene signatures in our YAP and TAZ refined signature lists using the fisher.test() function from the stats package (v 4.3.1). The p.adjust() function from the stats package (v 4.3.1) was used to calculate adjusted p-values for the multiple comparisons. R (v 4.2.2, 4.3.1 https://www.r-project.org/) code was implemented using RStudio (v2023.6 https://www.posit.co/).

## scRNA-seq analysis

Raw counts data were collected from GEO under accession GSE116237. Data were normalized and scaled using the 'LogNormalize' function with counts per 10,000 and the 'ScaleData' function, respectively. The

FindMarkers() function in Seurat (v4.3)[110] was used with the Likelihood-ratio test for single cell gene expression[111] to calculate differentially expressed genes between invasive (n = 41), neural crest stemm cell (n = 44), proliferative (n = 147), pigmented (n = 30), SMC (n = 224) and other (n = 188) cell states from Rambow et al.[12]. Violin plots were generated using the ggplot2 package (v3.4.3) (https://ggplot2.tidyverse.org/). R (v 4.3.1 https://www.r-project.org/) code was implemented using RStudio (v2023.6 https://www.posit.co/).

### Chemical synthesis of TEAD inhibitors

Synthesis of compounds 1-(7-(4-(trifluoromethyl)phenoxy)-3,4-dihydroisoquinolin-2(1H)-yl)prop-2-en-1-one (OPN-9643) was performed starting from commercially available material, *tert*-butyl 7-bromo-3,4-dihydro-1*H*-isoquinoline-2-carboxylate, which was reacted with 4,5,5-tetramethyl-2-(4,4,5,5-tetramethyl-1,3,2-dioxaborolan-2-yl)-1,3,2-dioxaborolane in presence of palladium catalyst, Pd(dppf)Cl$_2$ at 90 °C to obtain tert-butyl 7-(4,4,5,5-tetramethyl-1,3,2-dioxaborolan-2-yl)-3,4-dihydroisoquinoline-2(1H)-carboxylate (**3**) in 99% yield. Compound **3** was reacted with hydrogen peroxide in presence of sodium hydroxide to convert the boronate ester to the phenol derivative, tert-butyl 7-hydroxy-3,4-dihydroisoquinoline-2(1H)-carboxylate (**4**) in 84% yield. Compound **4** was then reacted with (4-(trifluoromethyl)phenyl)boronic acid in presence of a copper catalyst and oxygen to obtain the trifluorimethylphenyl ether derivative, tert-butyl 7-(4-(trifluoromethyl)phenoxy)-3,4 dihydroisoquinoline-2(1H)-carboxylate (**6**) in 74% yield. Compound **6** was treated with anhydrous hydrochloric acid in dichloromethane and 1,4-dioxane to remove the butoxycarbonyl group and to form the hydrochloride salt, 7-(4-(trifluoromethyl)phenoxy)-1,2,3,4-tetrahydroisoquinoline hydrochloride (**7**) which was then reacted with acryloyl chloride in presence of triethyl amine in dichloromethane at 0 °C and the crude product was purified by flash column chromatography using silica gel to obtain OPN-9643 in 20.78 g scale with 89% yield and > 99% purity.

Similar procedures were used to prepare 1-(7-(3-fluoro-4-(trifluoromethyl)phenoxy)-3,4-dihydroisoquinolin-2(1H)-yl)prop-2-en-1-one (OPN-9652). In this preparation, (4-(trifluoromethyl)phenyl)boronic acid was replaced by (3-fluoro-4-(trifluoromethyl)phenyl)boronic acid to react with intermediate **4** to obtain the intermediate, *tert*-Butyl 7-(3-fluoro-4-(trifluoromethyl)phenoxy)-3,4-dihydroisoquinoline-2(1*H*)-carboxylate (**10**), in 27% yield. Boc group removal from **10** followed by reaction with acryloyl chloride provided the final product, OPN-9652, in 18.83 g scale with 66% yield and > 99% purity. Further details of the synthesis procedures and characterization of OPN-9643 and OPN-9652 are provided in supporting information.

### Protein thermal shift assay

Depalmitoylated TEAD1 or TEAD4, or palmitoylated TEAD4 protein (0.5 μg/μl, approximately 23 μM) was incubated with 222 μM of OPN-9652 or OPN-9643, or with DMSO vehicle control in 25 mM HEPES pH 7.5, 150 mM NaCl, 0.01% Triton X-100 buffer for 1 hr. For depalmitoylated versus palmitoylated TEAD4 comparison, compounds are titrated from 222 μM top concentration in 8-point, 2-fold dose series. GloMelt Biotium dye was added before melt curve were measured on a QuantStudio 7 Flex. Delta Tm values are analyzed with a Protein Thermal Shift software (Thermo Fisher).

### Protein expression and purification

FLAG-tagged hTEAD1 and mTEAD4 were expressed in E. coli BL21 CodonPlus (DE3) RIPL cells (New England Biolabs, Inc.). Cells were grown in Terrific Broth media at 37 °C to an OD600 > 0.9, and protein expression was induced with 1 mM IPTG at 16 °C for 16-18 hrs. Cells were harvested via centrifugation and were lysed using a microfluidizer in 20 mM Tris pH 8.0, 200 mM NaCl, 20 mM imidazole, protease inhibitor cocktail, 0.2 mM PMSF. Lysate was cleared via centrifugation. Proteins were purified using Ni-NTA resin (Lifetech) in

20 mM Tris pH 8.0, 300 mM NaCl and eluted from the resin with 20 mM Tris pH 8.0, 300 mM NaCl, 300 mM imidazole. The Ni-NTA elution was applied to Superdex 200 26/60 column (Cytiva Life Sciences) that was equilibrated in 20 mM Tris pH 8.0, 100 mM NaCl. For preparation of depalmitoylated TEAD, proteins were diluted 1.5-fold with 1 M hydroxylamine hydrochloride (pH 8.0, 0.1% (2-hydroxypropyl)-B-cyclodextrin) and dialyzed against 1 M hydroxylamine hydrochloride (pH 8.0, 0.1% (2-hydroxypropyl)-B-cyclodextrin), 50 mM Tris pH 8.0, 100 mM NaCl, 2% glycerol for 4 hrs at room temperature. The protein sample was buffer exchanged into 20 mM Tris pH 8.0, 150 mM NaCl, 5% glycerol using a HiPrep 26/10 desalting column (Cytiva Life Sciences). Proteins were flashed frozen and stored at -80 °C for later use.

### Reporter assay

pLV-5XMCAT-luciferase-hPGK-Blast lentiviral reporter construct was transduced into A375 parental and SOX10 KO cell lines. Cells were plated and treated with OPN-9652 and OPN-9643 for 24 hrs, and cells were lysed in Passive Lysis Buffer (Promega), and Firefly luciferase activity was measured via Dual-Luciferase® Reporter Assay System using Glomax 20/20 luminometer. MSTO-211H cells were transfected to express a TEAD luciferase reporter and plated in growth media (RPMI-1640, 10% FBS, 1% Pen/Strep) in a black flat-bottom 96-well plate. The following day, medium was added with titrations of compounds in DMSO and cells were incubated in compound-containing medium for 24 hrs before viability (CellTiter-Fluor, Promega) and reporter (ONE-Glo, Promega) readout. Reporter luminescence signal was normalized to viability for individual wells for IC50 calculations. All experiments were performed in duplicate.

### Reverse phase protein array

Cells were plated in 6-well dishes at $3 \times 10^5$ cells per well overnight. On the next day, cells were washed twice in ice-cold PBS, and then lysed in 150 μL of RPPA lysis buffer (1% Triton X-100, 50 mmol/L Hepes (pH 7.4), 150 mmol/L NaCl, 1.5 mmol/L MgCl$_2$, 1 mmol/L EGTA, 100 mmol/L NaF, 10 mmol/L NaPPI, 1 mmol/L Na3VO4, 10% glycerol, protease and phosphatase inhibitors (Boehringer/Roche)) for 20 min with occasional shaking on ice. Lysates were centrifuged for 10 min at 18,000 x g, and the supernatants were collected. Protein concentration was determined by Bradford assay. Lysates were analyzed at the MD Anderson Functional Proteomic core facility (Houston, TX), where antibodies are extensively validated before being included in the panel. Serial dilutions of samples were arrayed on nitrocellulose-coated slides and run against 456 validated antibodies. A 3,3′-diaminobenzidine colorimetric reaction for a tyramide-based signal amplification approach was used to produce stained slides. The slides were scanned on a Huron Tissue-Scope scanner, and spot densities were determined using Array-Pro Analyzer. Relative protein levels were quantified using SuperCurve fitting and normalized for protein loading. BHFDR method was used to determine statistical significance. Apoptosis, ferroptosis, and cell cycle gene sets were collected from the Gene Ontology and KEGG Pathway databases. Phase-based cell cycle gene sets were collected from Whitfield et al. Statistical analyses were performed in (Matlab® v2022a).

### Statistical analysis

For the IncuCyte experiment comparing OPN-9652/OPN-9643 treatment +/- BRAFi + MEKi in A375 SOX10 KO cells, statistics were performed as follows. For each plate (replicate within treatment and cell line), the confluency at each time was first normalized by the confluency at time zero (0). The resulting ratios were linearly interpolated between time points, and the area under the resulting curves was computed for the entire time range of hrs elapsed. Log-transformed plate-specific AUCs were analyzed in two separate two-way ANOVA model with main effects of treatment and cell line and their

interaction. The first model was fitted to data from plates treated with combos including BRAFi + MEKi. The second model was fitted to the data from plates treated without BRAFi + MEKi. The interaction was retained in each model only if significant. The first model was used to estimate the mean difference in log AUCs (i) BRAFi + MEKi + OPN-9643 versus BRAFi + MEKi; (ii) BRAFi + MEKi + OPN-9652 versus BRAFi + MEKi. The second model was used to estimate the mean difference in log AUCs (i) between OPN-9643 and DMSO; (ii) between OPN-9652 and DMSO. Both models adjusted for the possible differences between the cell lines (A375.2.18 and A375.4.21). The analysis was performed in R (The R Foundation for Statistical Computing http://www.R-project.org).

## 3D spheroids

$5 \times 10^3$ cells/well were plated in a 96-well plate coated with 1.5% agarose and grown in suspension for five days to form spheroids. Briefly, spheroids were collected from 96-well plates using a transfer pipette and allowed to settle for 1 hr. A 24-well plate was coated with a collagen mixture (rat tail collagen type I containing reconstitution buffer, Ham's F-12 nutrient mix, and 5% FBS) and incubated for 20 min at 37 °C. Spheroids were resuspended in a collagen mixture, plated on the coated 24-well plates, allowed to set for 20 min at 37 °C, and covered with culture medium. For live/dead analysis, 3D tumor spheroids were stained with calcein-AM (7 μM, Invitrogen C1430) and propidium iodide (PI, 10 μg/mL) for live cells and necrotic/dead and pictures were taken with a C2 Nikon confocal Ti-Eclipse inverted microscope (Nikon) using NIS-Elements. Fluorescence of live/dead cells was quantified using ImageJ software, and the mean value (arbitrary units, AU) was used. For outgrowth area analysis, the Fluorescence mean AU value of calcein-AM was quantified using ImageJ software, and the spheroid core area was subtracted from the invasive outgrowth area.

## In vivo studies

Animal experiments were performed at a Thomas Jefferson University facility that is accredited by the Association for the Assessment and Accreditation of Laboratory Animal Care. The Institutional Animal Care and Use Committee at Thomas Jefferson University approved these studies (Protocol # 01052). All animals were provided with food and water *ad libitum* and housed in cages (with a maximum of 5 mice/cage) in a temperature and humidity-controlled environment. Animals were maintained in housing conditions that allowed for normal species behavior to minimize the development of abnormal behaviors and have access to humane and veterinary care. NOD.Cg-Prkdcscid Il2rgtm1Wjl/SzJ (NSG) mice were either purchased from Jackson Laboratories or bred in-house with IACUC approval. *BRAF* mutant A375 xenografts were generated in NSG mice, and when tumors were palpable, mice were treated with BRAFi + MEKi (PLX4720 200 PPM + PD-0325901 7 PPM) chow until MRD was reached (4 weeks). Mice were then randomly sorted into two cohorts: i) six mice (3 male and 3 female) were treated with BRAFi + MEKi chow; ii) 7 mice (3 male and 4 female) were treated continuously with BRAFi + MEKi and dosed twice per week with 50 mg/kg OPN-9652 via oral gavage. Three mice were excluded from the study due to severe weight loss or death due to fighting (one in the BRAFi + MEKi arm and two in the BRAFi + MEKi + TEADi continuous arm). *BRAF* mutant YUMM1.7 xenografts were generated in NSG mice, and when tumors were palpable, mice were treated with BRAFi + MEKi (PLX4720 200 PPM + PD-0325901 7 PPM) chow until tumors regressed (3 days). Mice were then randomly sorted into two cohorts: i) four mice (4 females) were treated with BRAFi + MEKi chow; ii) 5 mice (2 males and 3 females) were treated continuously with BRAFi + MEKi and dosed twice per week with 50 mg/kg OPN-9652 via oral gavage. One mouse was censored from the study due to unexplained death. Digital caliper measurements of the tumors were taken twice per week, and tumor volumes were calculated using the formula: volume (length × width²) × 0.52. Mouse weights were monitored twice

a week. The time to sacrifice for each animal was computed as the number of days for the tumor volume to exceed 1000 mm³, which does not exceed the maximal tumor burden of 1500 mm³ allowed by IACUC.

## Ethics statement

All research in this study complies with all relevant ethical regulations in accordance with IAUCAC, NIH, and Thomas Jefferson University.

## Reporting summary

Further information on research design is available in the Nature Portfolio Reporting Summary linked to this article.

## Data availability

The publicly available raw ATAC-seq data for melanoma cell lines used in this study are available in the SRA database under accession code SRP215051[39]. The publicly available RNA-seq data used in this study are available in the SRA database under accession codes SRP306463 for crSOX10 MeWo data, SRP329298 for A375 crSOX10 & CRT data, SRP329297 for 1205 LuTR PBRT data, and SRP029434 for A375 SOX10 knockdown data[18]. The publicly available scRNAseq data used in this study are available in the GEO database under accession code GSE116237[12]. The publicly available cell line gene dependence data used in this study are available on DepMap (https://doi.org/10.6084/m9.figshare.19700056.v2). The RNA-seq data generated for in this study have been deposited in the GEO database under accession codes GSE259388 (https://www.ncbi.nlm.nih.gov/geo/query/acc.cgi?acc= GSE259388) and GSE259389 for YAP1/TAZ knockdown and TEADi treatment in melanoma cells, respectively.

Crystal structures have been deposited at wwPDB under PDB accession code PDB 8S6Y. https://www.rcsb.org/structure/8S6Y. The remaining data are available within the Article, Supplementary Information or Source Data file. Source data are provided with this paper.

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

## Acknowledgements

We are grateful for support from the Sidney Kimmel Comprehensive Cancer Center Flow Cytometry, Translational Pathology, and Meta-Omics Shared Resources, which are supported by NIH/NCI Cancer Center Support Grant, P30 CA056036. At Thomas Jefferson University, we thank Dr. Maria Yolanda Covarrubias and Dr. Jason Hill for their help with confocal microscopy, Dr. Inna Chervoneva for her statistical expertise, and Frank Waltrich and Kristen DeRosa for tending to the mouse colony and breeding. We thank the Functional Proteomics Core Facility at The University of Texas, MD Anderson Cancer, and the Broad Institute for their help with RPPA and bulk RNA-sequencing, respectively. This work is supported by grants from the National Institutes of Health (NIH)/National Cancer Institute (NCI) R01 CA160495, R01 CA182635, and the Dr. Miriam and Sheldon G. Adelson Medical Research Foundation to A.E. Aplin. C.A. Ott is supported by the NIH/NCI award, F31 CA288084. C.Capparelli is supported by the Department of Defense (HT9425-23-MRP-MASA-ME230214) and the W.W. Smith Charitable Trust grants. Research reported in this publication utilized the Cancer Genomics and Flow Cytometry and Human Immune Monitoring Shared Resources at Sidney Kimmel Comprehensive Cancer Center, which are supported by the NCI Cancer Center Support Grant, P30 CA056036. The RPPA studies were performed at the Functional Proteomics Core Facility at The University of Texas MD Anderson Cancer Center, which is supported by the NCI Cancer Center Support Grant, P30 CA16672. The content is solely the responsibility of the authors and does not necessarily represent the official views of the NIH.

## Author contributions

C.A.O. conceived the study, designed the research, collected data, performed the analysis, and wrote the manuscript. T.J.P. collected data and performed the analysis. P.Y.C. collected data and performed the analysis. S.C. collected data and performed the analysis. G.M. collected data and performed the analysis. K.L. collected data and performed the analysis. G.L.M. collected data and performed the analysis. M.T. collected data and performed the analysis. W.D.M. collected data and performed the analysis. S.D.V. collected data and performed the analysis. D.A.E. helped with manuscript revisions. J.L. helped design the manuscript, provided technical support, and made manuscript revisions. C.C. helped design the manuscript, prepared figures, and manuscript revisions. G.B. helped design the manuscript, provided technical support, and made manuscript revisions. A.E.A. conceived the study, designed the research, and wrote the manuscript.

## Competing interests

A.E. Aplin has an ownership interest in patent number 9880150. Pan-Yu Chen, Somenath Chowdhury, and Gideon Bollag are employees of Opna-Bio. The other authors declare no competing interests.
