## [Transparent Peer review file · Nature Communications]

Targeting TAZ-TEAD in minimal residual disease enhances the duration of targeted therapy in melanoma models

Corresponding Author: Professor Andrew Aplin

Version 0:

Reviewer comments:

Reviewer #1

(Remarks to the Author)

In this manuscript, Ott et al showed that TEAD-TAZ signaling mediates the tolerance to BRAFi and MEKi in melanoma, which represents a druggable vulnerability. The authors demonstrated that melanoma with resistance to MAPK-targeted therapies exhibited loss of SOX10, with concurrent induction of TEAD-TAZ activities. In addition, new TEAD inhibitors were developed in this study. In vivo drug resistance model showed that combining TEADi and MAPKi improved therapeutic efficacy. The studies are interesting and potentially could be impactful for melanoma. However, the underlying mechanisms by which TEAD-TAZ mediates drug resistance in SOX10-deficient melanoma are still unclear. How SOX10 upregulates TEAD-TAZ signaling needs to be explored. Although Merlin/NF2, the upstream regulator of Hippo pathway, is downregulated in SOX10 KO A375 cells, it is not consistent in SOX10 KO MeWo cells, and TEAD-TAZ signaling is activated in both systems, suggesting that merlin might not be the key point in SOX10 KO-induced activation of TEAD-TAZ signaling. Lack of understanding of mode of action will hinder the therapeutic application in the future.

Several issues should be addressed before its consideration for publication in Nature Communication:

Major:

1. Although TAZ S89A could promote tolerance to BRAFi and MEKi in A375 parental cells, whether TAZ is indispensable in the resistance to MAPK pathway inhibitors in SOX10-deficient melanoma required more evidence. For example, whether knocking out TAZ could rescue the tolerance in SOX10-deficient melanoma? Loss of TAZ might also induce YAP expression as a compensation mechanism, and it might require KO of TAZ and YAP.
2. If TAZ induces drug tolerance in a TEAD-dependent manner, knocking out TEAD in A375 parental cells would rescue the effects induced by TAZ S89A. Therefore, to confirm TEAD works together with TAZ to mediate drug resistance, the authors could test if TEAD KO rescues the tolerance to BRAFi and MEKi in A375 cells overexpressing TAZ S89A.
3. Figure 1E showed that SOX10 KO in A375 and MeWo promoted TEAD transcriptional activities. Did the authors detect whether the expression levels of YAP, TAZ and TEAD are increased? How does SOX10 suppress TEAD-TAZ activation?
4. To confirm that OPN-9643 and OPN-9652 target the palmitoylation pocket of TEAD proteins, the authors are suggested to test if these two compounds could inhibit TEAD palmitoylation in vitro. How does this compound compare to K975 and other covalent inhibitors? What makes these new compounds more attractive (TEAD isoform selectivity? selectivity over other targets?)
5. Figure 4A showed the predicted dependency to YAP and TAZ across 62 melanoma cell lines. Whether the dependency of TAZ is correlated with SOX10 deficiency? Whether the five melanoma cell lines with TAZ dependency, including WM983B, are SOX10-deficient?
6. In Figure 7D, the authors claimed that the triple combination significantly reduced outgrowth area compared to DMSO and TEADi alone conditions in both 2.18 and 4.21 cells. However, it seems like that there is no statistically significant difference between BRAFi plus MEKi and triple combination, suggesting that TEADi did not sensitize SOX10 KO A375 to BRAFi and MEKi in this assay.
7. In the left panel in Figure 7E, is the PI intensity in triple combination group significantly higher than that in BRAFi plus MEKi group? If it is not statistically significant, it might be not appropriate to conclude that triple treatment has synergistic effects on cell death. Bliss scores which usually used to evaluate synergy should be used.

Minor:

1. Fig. 1A did not show the mesenchymal gene signature for MeWo cells, which is not consistent with the figure legend.
2. Figure 3B showed that hyperactive TAZ S89A more efficiently induced TEAD target genes than YAP1 S127A. The authors should show comparable expression levels of TAZ S89A and YAP1 S127A. In addition, it seems like that there is no

significant difference between TAZ S89A and YAP1 S127A in up-regulating CYR61, though the effect on CTGF was dramatic.

3. In line 179, there should be a space between "TEAD1-Y421E" and "in A375 SOX10 KO cells".

4. The statistical significance should be shown in Figure 7A.

5. Does the effects on CTGF and CYR61 expression in CRT cells could be phenocopied in PBRT models? These results will provide stronger support for the effects of OPN-9652 and OPN-9643 in acquired-resistant models.

6. The numbers of mice used in in vivo study should be mentioned.

7. "auto palmitoylation" should be replaced by "autopalmitoylation".

8. The methods for purification of depalmitoylated TEAD protein should be given (or referenced).

Reviewer #2

(Remarks to the Author)

In this study, Ott et al describe novel TEAD inhibitors that act on resistance to BRAFi inhibitors and block emergence of BRAFi-resistant cells responsible for tumour regrowth.

The following issues should be addressed.

1. The authors use A375 cells as a model and they have generated SOX10 KO derivatives that then undergo a pseudo-EMT to acquire a more mesenchymal phenotype. The authors should indicate that A375 are MITF-low NCSC like cells. They reproduce many of the effects in MEWO cells with SOX10 KO. MEWO cells are much more melanocytic and express MITF. The fact that SOX10 KO in both leads to a comparable mesenchymal TEAD and TAZ phenotype should be mentioned.

2. Figure 2C is likely to be very difficult to read, can the authors think of a clearer way to present the data. A simple Z-scored heat map would probably suffice.

3. In the first figures I could find no indication of the expression levels of TEAD1 and TEAD4 or YAP and TAZ between the parental and SOX10 KO A375 cells. Clearly, TEAD-TAZ-dependent transcriptional activity is induced by SOX10 KO, but what really happens at the protein level for these factors. The authors mention (line 116) that the underlying mechanism is not known. Immunoblots (showing up-regulation) and immunofluorescence (presence of TEAD and TAZ in the nucleus) would allow the authors to make a stronger statement on this. Blots of parental cells are not shown either in Figure S1.

4 The authors develop novel inhibitors that clearly block TEAD-TAZ-dependent gene expression. However, there is no mechanistic insight into these effects. The authors claim the compounds target the palmitoylation pocket, but what are the consequences for TEAD proteins in cellulo, are they destabilized, do they no longer interact with TAZ, are they nuclear or cytoplasmic. Additional experiments that address these issues are essential to strengthen the authors hypothesis.

5. The authors use PD0325901 and PLX4720 as MEKi and BRAFi. However, my understanding is that Dabrafenib and trametinib are the clinically approved and most used combinations. Can the authors comment on this?

6. In the last section of the results, the authors say that they have previously used PDX models that progress to MRD under MAPKi treatment (lines 292-294). However, they then go on to use A375 CDX as a model for the preclinical validation of the novel inhibitors. Why did the authors not use a PDX model for this critical validation experiment? This referee would very strongly suggest that the TEADi be tested with a PDX model, rather than a CDX model. Alternatively, at least the use of multiple CDX models would give a better idea of the general applicability of the use if these inhibitors. Moreover, the number of mice in each arm is shown neither in the figure, the legend nor the materials and methods. Looking at the Kaplan-Meier in panel 8I one has the impression from the number of events that only a 3-4 mice are in the final cohort. This is largely insufficient at least 10 mice should be used in each arm. Can the authors clarify this?

7. Lastly, can the authors benchmark these new inhibitors compared to those previously described and in current clinical trial? What would be the advantage of the new ones over those already characterized in particular IAG933.

Reviewer #3

(Remarks to the Author)

This work focuses on the study of drug tolerance during acquisition of melanoma resistance to BRAF and MEK inhibitors in melanoma, in the setting of SOX10 loss and its relationships to YAP/TAZ-dependent regulation of TEAD pathway. Authors describe that SOX10-deficient melanoma cells display upregulation of the TEAD pathway, as measured by induction of the TEAD targets CTGF and CYR61, through a predominant TAZ-dependent mechanism. They have generated and used novel TEAD inhibitors to show re-sensitization of these cells to BRAF and MEK inhibitors. They propose that these inhibitors could potentially be used for treatment of melanoma before acquisition of resistance to targeted therapies.

This is an interesting, significant and potentially relevant study in the field of cancer resistance, that might lead to future experimental research to target the onset of acquired resistance to targeted therapies in melanoma, in addition of the current clinical trials with TEAD inhibitors for mesothelioma. The manuscript is well written and the methodology is correctly presented.

At present, the manuscript has several limitations that overall reduce the quality of the study. The main one is already

mentioned by the authors on line 116: "Mechanistically, how SOX10 loss leads to up-regulation of TEAD targets is not known." Adding experimental work on this issue will enhance the manuscript. Another point that must be considered is that the study focuses on SOX10-deficient melanoma cells tolerant/resistant to MAPKi, which is clearly demonstrated here, but that it could represent a minority of the mechanisms leading to tolerance. Also, the study is performed with only one SOX10-silenced BRAF-mutant melanoma cell line (although two different clones), in addition of the BRAF wt Mewo melanoma cells. It will be important to include an additional BRAF-mutant melanoma cell line to have a more robust output. The study lacks experimental data showing YAP/TAZ translocation to nucleus, a step required for TEAD activity. Linked to this point, does SOX10 depletion leads to dephosphorylation of YAP/TAZ in the cells used in this study?

Measuring cell growth by analyzing cell culture confluency does not seem the most appropriate method, as some cells in early or advanced apoptosis might be included in this measure. In addition, authors must give more convincing data on the specificity of the TEAD inhibitors, by including additional controls. Authors have limited their TEAD-dependent gene signature to CTGF and CYR61; however, DKK1, WNT5A, TGF-beta2, MHYC and GLUT3 are also TEAD targets. Moreover, it is known that TAZ overexpression upregulates PD-L1, and YAP/TAZ regulates TGF-beta signalling, adding more complexity to the system. In fact, EGFR- and TGF-beta-dependent signaling are intertwined with YAP/TAZ signaling. Finally, some of the data shown on the figures lack some controls, and the flow of the text and some of the figures must be corrected.

The study will also improve if reported molecular players associated with tolerance are included. For instance, tolerance has been linked to increased NGFR and KDM5B, and to decreased Melan A expression. Furthermore, it will be interesting to analyze MITF, a downstream target of SOX10 which could have some links to the TEAD signaling.

Specific points

- Fig. 1.

Panel A. Figure legend mentions MeWo cells which do not appear there.

Panel B. What do the data mean in terms of SOX10 downregulation?

Panel C. Significance is especially strong in Mewo cells, and modest in 2.18 and 4.21 cells.

Panel E. Expression of YAP and TAZ is missing, although appearing on Fig. 3. It will be important to add it here to better link to induction of CTGF and CYR61. These data will also improve by showing whether re-introduction of SOX10 in the 2.18 and 4.21 cells reverses induction of CTGF and CYR61.

Panel F: Does SOX10 regulates Merlin? Comment on this.

Line 112: A reference is missing.

Fig. 2.

Panel A. Text mentions YAP1-knockdown and TAZ-knockdown, but proof of silencing does not show up until Fig. 3A. It will be better to add it here.

Panel D, step 3. Correct "data" (instead of sata).

Panel E. Which siRNA have been used, 1 or 2?

Line 138. Text mentions. "We observed a greater enrichment in all 4 gene sets following TAZ..." In fact, data show enrichment in one set, a tendency in another, and no enrichment in the other two.

Line 139. Changing suggesting for demonstrating will be more appropriate.

Line 160. Add "A375" in SOX10-deficient cells.

Fig. 3.

Panel A. It will be important to show YAP and TAZ expression in parental A375 cells for comparison.

Panel B. As indicated above, it will be important to show the YAP/TAZ translocation to nucleus in cells expressing the active YAP/TAZ forms. Figure legend does not mention how many times was this experiment performed. Can authors also show data on Mewo cells?

Panel C. As indicated above, it would be better to measure tolerance with another type of growth analysis. Also, parental A375 cells seem not to be affected by BRAFi+MEKi. Can authors explain this fact?

Panels D, E. Data will improve by adding parental A375 for comparison. 4.21 cells display a clear reduction in pErk1/2 upon treatment with BRAFi+MEKi, but their growth seems quite similar to control cells. Any comment? The p values are a bit confusing the way they are shown on panel E.

Figure 4.

Panel C. Must show SOX10 expression along the lanes.

Panel D. See comment on Panel C of Fig. 3.

Fig. 5.

Why/How did they choose these molecules to inhibit the TEAD targets? (only based in the structure of cysteine residues or by any other screening experiment?)

The data will definitely improve if adding a non-inhibiting control from the screens. In lines 208-209 appears compound OPN-9651, but is not longer mentioned or shown in the figure. Is it a negative control?

Line 212. Missing a reference.

Lines 212-215. Data on dose response not shown. Only results in Fig. 5C. Could authors show it in a supplemental figure?

Fig. 6.

Panel A. Separation of data in three panels does not allow to visualize whether increased luciferase activity especially in 2.18 cells is statistically significant compared to parental cells in the absence of both inhibitors.

Panels A-C: Show data also using an inactive TEAD inhibitor as control

Panels B, C: If possible, show expression of a non-TEAD target (in addition of the HSP90 loading control).

Panel D. p value? One replicate? Concentration of inhibitors of 2 M?

Panel E. In order to show specificity, and to increase the observation of the direct relationships between the observed alteration and TEAD inhibition, can you compete TEAD inhibitor with expression of active TAZ to check reversion (or reduction) of some of the targets?

Fig. 7.

Panel A. Lacking parental A375 cells

Panel C. Is OPN-9643 exerting similar effects of the one displayed by OPN-9652?

Panel F. Missing control Dox alone

Fig. 8.

Panel B. Enrichment plot for PBRT resistant cells (from 1205Lu treated with BRAFi/MEKi)?

Panels C-E. Different times used for the analyses. Any comment?

Panel D. siTAZ2 seems to also silence YAP, and siYAP2 also silences TAZ? Any explanation?

Panel F. Immunoblotting shows data from total tumor. The possible contribution of the tumor microenvironment must be commented. Also, can you also show data on CTGF?

Version 1:

Reviewer comments:

Reviewer #1

(Remarks to the Author)

The authors have greatly improved the manuscript with the revision. All of my previous comments have been addressed. I support the publication of the paper.

Reviewer #2

(Remarks to the Author)

In the revised version, the authors have added novel data that satisfactorily addresses most of the issues raised by this referee.

However, the novel data on the expression and localization of YAP/TAZ raise new questions. In point 3 of my original review (this issue was raised by the other referees), I asked about the expression and localization of YAP/TAZ and the TEAD factors. The authors indicate in their reply that the expression of these factors is not in fact significantly altered by SOX10 KO. This begs the question then of the activity of these factors in the SOX10-expressing cells, presumably they are not 'active' and hence how does the expression of SOX10 repress or bypass their activity. Given the new data, this becomes a central question of the manuscript (see lines 373-379). As mentioned before, the development of the novel TEAD inhibitors is an important and novel development for the field, but the rest of the data showing the activity of the TEAD/TAZ pathway in the more mesenchymal state is much less novel. Given that the authors state that there are no changes in TEAD/TAZ expression/localization upon SOX10 KO, the question of the cross-talk between these pathways becomes a major issue that has not really been addressed. What is the effect for example of TEAD or TAZ silencing in the SOX10-expressing parental lines, what signaling pathways are changed upon SOX10 KO, does the phosphorylation state of TAZ or TEAD change; once activated can the activity of the TEAD/TAZ pathway be repressed by SOX10 re-expression.

Another minor outstanding issue is the very small number of mice used for the CDX experiments (6 and 7 in one group and 4-5 in the other). This really at the limit of statistical significance, perhaps the authors could add a supplemental Figure showing the growth of each individual tumour in each arm as well as the aggregate shown in the main figure.

Reviewer #3

(Remarks to the Author)

Authors have addressed the comments raised by this reviewer. Some of the new data reasonably answer the comments, but some other remain unanswered, or even addressed without taking into account this reviewer's comments.

Through the text, authors use the terms tolerance and growth without a clear distinction at given experimental contexts. This must be clarified.

Discussion lacks a conceptual approach of what the results of this study provide for treatment of melanoma cells in the context of low/absence of Sox10.

Specific points

- The previous revision mentioned that this study will be strengthened by inclusion of data with at least an additional BRAF-mutant melanoma cell line silenced for SOX10.

The authors have used the mouse melanoma cell line YUMM 1.7 (BRAF-mutant) directly in xenograft studies using the triple inhibitor combination as shown in the final part of the manuscript. Although they show that median survival increases statistically significantly with this combination, data is inconclusive, as they are missing the links between SOX10 absence,

YAP/TAZ and TEAD. This must be addressed.

- Another comment from the first revision related to the lack of experimental evidence regarding YAP/TAZ translocation to nucleus, a step required for TEAD activity. Also it was asked to test if SOX10 depletion leads to dephosphorylation of YAP/TAZ in the cells used in this study.

Authors found no significant differences in TAZ nuclear localization between SOX10 KO cells and parental controls (new Supplemental Fig. 3C). Furthermore, they assessed the phosphorylation status of YAP at three key LATS1/2 mediated phosphorylation sites and observed no changes in phosphorylation between parental and knockout cells (new Supplemental Fig. 1C). In addition, this supplementary figure shows increased YAP phosphorylation in SOX10 KD cells, and no data of TAZ phosphorylation is displayed. Also, enhanced YAP could induce its proteasomal-dependent degradation, which is opposite to what is shown on Supplemental Fig. 1C. The data also remains inconclusive, as TAZ and YAP should translocate to nucleus once they are dephosphorylated to activate TEAD.

- To measure cell growth by analyzing cell culture confluency.

Although authors have addressed the measurement of cell growth and apoptosis in the context of using the OPN inhibitors with MeWo SOX10 lo (new fig. 7C), this reviewer still considers that looking at cell confluency to define drug-tolerant persisters might not take into account issues such as cell shape and spreading, and cell-cell packing.

- Analysis of the expression of molecular players associated with tolerance, including NGFR, KDM5B and Melan A. Also checking the levels of MITF, a downstream target of SOX10 which could have some links to the TEAD signaling.

Authors show in Supplemental Fig. 1A the expression of these markers in the context of drug tolerance using the single-cell RNA-seq dataset from Rambow et al. on minimal residual disease. Although interesting, they have not tested the levels of these markers in their SOX10 KD cells, which will make the data more convincing.

- Previous line 160.

As indicated by authors, text has been amended to ... "suggest that TAZ is the major co-activator of the TEAD transcriptome in cutaneous melanoma cells"....

I would suggest to change the last part of the sentence to "...of the TEAD transcriptome in SOX10 low cutaneous melanoma cells"

- Comment on old Fig. 3B. Translocation of YAP/TAZ to nucleus.

In the response to this reviewer, authors have performed immunofluorescence for HA-tagged YAP and TAZ constructs with point mutations preventing phosphorylation and cytoplasmic sequestration. In their response Fig. 5 they display data on A375 YAP-S127A cells showing that the active YAP form is localized in the nucleus. That looks fine, but author's data conclude (subheading, line 121) that "TAZ is a major regulator of TEAD targets in SOX10-deficient melanoma cells." Therefore, it will be important to show the new data of activated TAZ (S89A mutant) in a figure, not in a Response figure. Furthermore, in the blot of the new Supplemental Fig. 3B they show HA-tag expression in TAZ S89A but not in YAP S127A. This is contradictory to what authors display in response Fig. 5.

Also, on this old Fig. 3B (now new fig. 3F), there is no HA label on the blot corresponding to the YAP S127A cells. Reason?

- Comment on old Fig. 3D, 3E.

Unless I missed in the new version, I can find comparison with parental A375 cells.

- Comment on old Fig. 4C.

The data is now shown in Fig. 3D, which displays SOX10 expression in WM983B cells independently of high levels of the YAP/TAZ/TEAD targets CTGF and CYR61. The authors mention in the revised version, lines 177-180: "...There was no correlation between SOX10 expression and predicted YAP1 or TAZ dependency; however, there was a correlation between SOX10 expression and TEAD1 dependency (Supplemental Fig. 3A) and between predicted TAZ dependency and predicted TEAD1 dependency (Fig. 3C)." Yet, data on Figs. 1 and 2 with SOX10 KD A375 cells suggest a pathway linking SOX10 with YAP/TAZ and TEAD. Later, in lines 199-202, authors mention: "...Despite there being no correlation between SOX10 expression and YAP1 or TAZ dependency at steady-state, further analysis is warranted to determine mechanisms that TAZ drives a SOX10-deficient drug-tolerant phenotype." What authors mean for steady-state? A clear explanation for these discrepancies must be provided.

- Comment on old Fig. 5.

The comment related to the use of a non-inhibiting TEAD inhibitor. In the response Fig. 6, authors show data using several TEAD inhibitors of the VT group. They mention that VT106, a 50x less potent enantiomer of VT104, displays a lower reduction in luciferase expression. How is this inhibition in comparison with the OPN inhibitors in the same, not separated experiments? I understand that sometimes it is difficult to find the appropriate control, but the above mentioned comparison might provide more useful information.

In the previous version of the manuscript, the IC50 for NCI-H226 was 3 μ M in Fig. 5C. Now it shows 100 nM. Please, explain this change. Also in this panel, the IC50 MSTO-211 TEAD values do not match to what is written in the text.

- New Fig. 6A.

Points in the A375 4.21 data bars are missing.

Version 2:

Reviewer comments:

Reviewer #2

(Remarks to the Author)

The authors have addressed my major comments by showing that in fact its not TEAD and YAP per se that are subject to regulation by SOX10 KO, but AP1 through up-regulation of cJUN expression. Increased cJUN and AP1 activity is seen with EMT in melanoma cells as previous data have shown that the AP1-TEAD module is the key component of the mesenchymal enhancers. This explanation therefore seems coherent with the findings of this paper. The authors have gone to some length to address all of these issues and I can now recommend publication.

Reviewer #3

(Remarks to the Author)

Authors have addressed the new comments raised by this reviewer. The manuscript now shows a clear improvement, especially on data from Figs. 5-8 and associated supplemental figures, which constitute an interesting characterization of the TEAD inhibitors. However, there are still few pending points that need to be addressed. A main pending concern is that induction of TEAD targets seems not to be always associated with loss or reduction of Sox10.

Specific points

- Fig. 1D: If TAZ is similarly expressed in parental and Sox10-depleted cells (see suppl. Fig. 1D), why there is no induction of CTGF and CYR61 expression in parental cells? Is it possible that the presence of Sox10 blocks the induction?

- Suppl. Fig. 1D, E: Authors mention a modest increase in YAP, TEAD1 and panTEAD (lines 127-130). However, the western blots do show remarkable increases. Have authors quantified the blots? Authors must consider to show data from this supplementary figure as part of main Fig. 1, to have data in a more logical and ordered form. Otherwise, the data looks a bit disorganized.

In this same figure (panel D), phosphorylation status of YAP1 and TAZ is shown, but data is not commented in the text.

- Sentences within lines 146-150 show some repetition. Must clarify

- Line 173 should read: ...melanoma cells depleted of Sox10 for mediating...

- Fig. 3D: WM983B cells express Sox10, TAZ, YAP1 and also CTGF and CYR61. Therefore, these cells should have enhanced TEAD-dependent signaling in the presence of Sox10. This is different from data shown on fig. 1D.

- Line 219: This new sentence should read:...primary TEAD co-activator in Sox10-depleted melanoma, its requirement...

- Fig. 3F: How do you rule out that there is not too much effect of mutant YAP1 due to the fact that it might not translocate to nucleus? Authors show no data on YAP translocation to nucleus. Text related to this figure does not mention the YAP1 data shown in the figure.

- Lines 250-253: As there are no differences in TAZ expression between parental A375 and Sox10-knockdown cells, would TAZ depletion in parental A375 cells also lead to decreased growth? If so, this would suggest no correlation between TAZ and Sox10.

- Fig. 4D, lines 258-259: Unless I am confused, there are no differences in percent confluency between induced and non-induced TAZ-S89A. Therefore, the sentence is unclear.

- Fig. 6A, lines 310-313: Data suggest that inhibition is independent of the presence or absence of Sox10

- Suppl. Fig. 5F, lines 315-317: Again, independently of Sox10

- New text in lines 324-327: There seems to be no conclusion from data in suppl. Fig. 6A. Are VT106 and VT107 also good TEAD inhibitors? Also in the Rebuttal letter, on Comment on old Fig. 5, there appear to be a confusion between VT104 and VT107.

- Suppl. Fig. 9A-D: On one hand, authors use the Sox10-expressing A375 in NSG models. On the other hand, panels B-D display data using Sox10-deficient cells. As data show similar results, it is suggested that effects are independent of the presence or absence of Sox10.

- Line 429, Discussion: I think using "Mechanistically" here is going too far. Authors must soften this sentence.

RESPONSE TO REVIEWERS (Ott et al.)

We appreciate the reviewers' feedback, which has helped us to strengthen our manuscript. We have expanded our results to include additional MeWo cell line models. We demonstrate that the induction of YAP1 S127A and TAZ S89A is sufficient to promote tolerance to MEK inhibitor (MEKi) in MeWo parental cells (Supplementary Fig. 3D). We also determined that TEAD inhibitor (TEADi) treatment sensitizes MeWo SOX10 knockout (KO) cells to MEKi (Fig. 7B) and induces cell death markers in MeWo SOX10 KO cells (Fig. 7C). Furthermore, we have added an additional *in vivo* model to further strengthen the evidence of efficacy of combinatorial therapies with TEADi (Supplemental Fig. 7A).

A major effort in the revised manuscript was to characterize the TEADi at a structural level and in terms of transcriptional readouts. We have performed co-crystallization studies of TEAD1 and OPN-9652 (Fig. 5D), conducted RNA-seq analysis following treatment of the TEADi in A375 cells (Fig. 6F and Supplementary Fig. 5A), and compared these inhibitors to other company's compounds to provide a broader context for the efficacy of the novel inhibitors that are described in our work (Figure 1, and Figure 6, below).

Below, we address each reviewer's comments in detail.

REVIEWER COMMENTS

Reviewer #1 - Drug development, TEAD (Remarks to the Author):

The mechanisms by which TEAD-TAZ mediates drug resistance in SOX10-deficient melanoma are still unclear. How SOX10 upregulates TEAD-TAZ signaling needs to be explored. Although Merlin/NF2, the upstream regulator of Hippo pathway, is downregulated in SOX10 KO A375 cells, it is not consistent in SOX10 KO MeWo cells, and TEAD-TAZ signaling is activated in both systems, suggesting that merlin might not be the key point in SOX10 KO-induced activation of TEAD-TAZ signaling. Lack of understanding of mode of action will hinder the therapeutic application in the future.

Several issues should be addressed before its consideration for publication in Nature Communication:

Major:

1. Although TAZ S89A could promote tolerance to BRAFi and MEKi in A375 parental cells, whether TAZ is indispensable in the resistance to MAPK pathway inhibitors in SOX10-deficient melanoma required more evidence. Loss of TAZ might also induces YAP expression as a compensation mechanism, and it might requires KO of TAZ and YAP.

In response, we knocked down TAZ in SOX10KO drug-tolerant cells and showed that the depletion of TAZ reduces BRAFi/MEKi tolerance in these cells. These new data are included in Fig. 4B. TAZ knockdown does not alter YAP1 expression (Fig. 2A). These data show that TAZ is the main paralog required for drug tolerance in SOX10-deficient melanoma, and that loss of TAZ doesn't induce YAP expression in melanoma.

2. If TAZ induces drug tolerance in a TEAD-dependent manner, knocking out TEAD in A375 parental cells would rescue the effects induced by TAZ S89A. Therefore, to confirm TEAD works together with TAZ to mediate drug resistance, the authors could test if TEAD KO rescues the tolerance to BRAFi and MEKi in A375 cells overexpressing TAZ S89A.

We used a combination of 4 TEAD siRNA sequences to knock down TEAD expression in A375 TAZ-S89A cells. Pan knockdown of TEADs reverses the drug tolerance induced by overexpressing TAZ-S89A, indicating that TAZ-S89A mediates drug tolerance in a TEAD-dependent manner. These new data have been added to Figures 4C and 4D. Lines 217-221

3. Figure 1E showed that SOX10 KO in A375 and MeWO promoted TEAD transcriptional activities. Did the authors detect whether the expression levels of YAP, TAZ and TEAD are increased? How does SOX10 suppresses TEAD-TAZ activation?

We detected a modest increase in YAP1, TEAD1, and pan-TEAD expression in A375 SOX10 KO cells compared to parental, but there were no changes in TAZ or other TEAD paralog expression (Supplemental Fig. 1C-D). There were no changes in YAP1, TAZ, or TEAD expression between MeWo parentals and MeWo SOX10 KO cells (Supplemental Fig. 1C-D). Furthermore, TAZ localization via IF does not show a difference between A375 Parental and SOX10 KO cells (Supplemental Fig. 3C). Thus, we believe that there are multiple mechanisms that contribute to the upregulation of TEAD activity in SOX10 KO cells.

4. To confirm that OPN-9643 and OPN-9652 target the palmitoylation pocket of TEAD proteins, the authors are suggested to test if these two compounds could inhibit TEAD palmitoylation *in vitro*. How does this compound compared to K975 and other covalent inhibitors? What makes these new compounds more attractive (TEAD isoform selectivity? selectivity over other targets?)

To confirm the specificity of these novel TEAD inhibitors, we performed co-crystallization of OPN-9652 with TEAD1. We show that OPN-9652 covalently binds cysteine 359 within the central hydrophobic core of TEAD1 (Fig. 5D). These data complement the thermal shift of OPN-9652/OPN-9643 with palmitoylated and depalmitoylated TEAD4 (Supplemental Figure 4C).

K975 was initially used in the project as a reference, but it does not have good pharmacological properties *in vivo*. However, Vivace Therapeutics allowed us to compare our inhibitor OPN-9652 with their inhibitors, VT103 and VT104. In A375 4.21 TEAD luciferase reporter cells (Response Fig. 1A) and through western blot analysis of CTGF and CYR61 expression (Response Fig. 1B), OPN-9652 and VT103 have similar inhibition levels and are more potent than VT104. The main difference between OPN-9652 and Vivace compounds is that OPN-9652 is a covalent inhibitor.

Response Figure 1. A A375 crSOX10 #4.21 cells were treated with 0.5 μM of either OPN-9652, VT103, or VT104 for 24 hrs and then cells were lysed. Firefly luciferase activity was measured via Dual-Luciferase[®] Reporter Assay System. The experiment was repeated three times with similar results. **** $p < 0.0001$, One-way ANOVA. **B** A375 crSOX10 #4.21 cells were treated with either 0.5 μM OPN-9652, 0.5 μM VT103, or 0.5 μM VT104 or

vehicle control, for 24 hrs. Cell lysates were analyzed by Western blotting with the antibodies indicated.

5. Figure 4A showed the predicted dependency on YAP and TAZ across 62 melanoma cell lines. Whether the dependency of TAZ is correlated with SOX10 deficiency? Whether the five melanoma cell lines with TAZ dependency, including WM983B, are SOX10-deficient?

We further examined DepMap data and show there is no correlation between TAZ and SOX10 expression (Supplemental Fig.3A); however, it should be noted that the dependency of TAZ and SOX10 is in the absence of BRAFi + MEKi. We next explored TEAD1 dependency and SOX10 expression and found that there is a correlation between TEAD1 dependency and SOX10 expression (Supplemental Fig. 3A). There is also a positive correlation between TEAD1 dependency and TAZ dependency (Fig. 3C). WM983B cells express SOX10, and SOX10 expression remains unchanged following the knockdown of YAP and TAZ (Fig. 3D).

6. In Figure 7D, the authors claimed that the triple combination significantly reduced outgrowth area compared to DMSO and TEADi alone conditions in both 2.18 and 4.21 cells. However, it seems like that there is no statistically significant difference between BRAFi plus MEKi and triple combination, suggesting that TEADi did not sensitize SOX10 KO A375 to BRAFi and MEKi in this assay.

We agree with the reviewer that there is indeed no statistically significant difference between the combo and triple therapy in our spheroid experiment. In the revised manuscript, we more clearly state the effects (Lines 321-323).

7. In the left panel in Figure 7E, Is the PI intensity in triple combination group significantly higher than that in BRAFi plus MEKi group? If it is not statistically significant, it might be not appropriate to conclude that triple treatment has synergetic effects on cell death. Bliss scores which usually used to evaluate synergy should be used.

We agree with the reviewer that there was not a significant difference in PI intensity in the 2.18 cells between BRAFi + MEKi and the triple combination. Additionally, we probed for the expression of cell death markers (GSDME, PARP) in MeWo cells in 2d and saw an increase in expression in those proteins (Fig. 7C), which strengthens our interpretation.

Minor:

1. Fig. 1A did not show the mesenchymal gene signature for MeWo cells, which is not consistent with the figure legend.

We now include MeWo enrichment plots in Fig. 1A. Thank you for bringing this to our attention.

2. Figure 3B showed that hyperactive TAZ S89A more efficiently induced TEAD target genes than YAP1 S127A. The authors should show comparable expression levels of TAZ S89A and YAP1 S127A. In addition, it seems like that there is no significant difference between TAZ S89A and YAP1 S127A in up-regulating CYR61, though the effect on CTGF was dramatic.

We have amended the text to reflect that there is a modest increase in CYR61 expression and a dramatic effect on CTGF expression (Lines 193-194.). We have further transduced hyperactive TAZ S89A and YAP1 S127A into MeWo parental cells (Supplemental Fig. 3B)

and show again that there is a modest increase in CYR61 expression and a large increase in CTGF expression following the induction of TAZ S89A.

3. In line 179, there should be a space between “TEAD1-Y421E” and “in A375 SOX10 KO cells”.

Fixed. Thank you for the correction.

4. The statistical significance should be shown in Figure 7A.

The statistical analysis has been moved to Supplemental Fig. 6B, but we have also noted the significance on the graph in Fig. 7A and new data in Fig. 7B

5. Does the effects on CTGF and CYR61 expression in CRT cells could be phenocopied in PBRT models? These results will provide stronger support for the effects of OPN-9652 and OPN-9643 in acquired-resistant models.

RNA-seq of publicly available YAP/TAZ gene signatures implies an enrichment in YAP/TAZ signaling in the PBRT cells. We probed for CTGF, CYR61, and AXL in the PBRT lysates, and we detected an upregulation of CYR61, CTGF and AXL in PBRT16. The findings are less consistent for PBRT15, with only CYR61 being upregulated (Response Fig. 2).

Response Figure 2. 1205 parental, PBRT15, and PBRT16 cells were plated and left untreated or treated with 1 μ M of PLX8394 (Paradox breaker BRAF inhibitor). Cell lysates were analyzed by Western blotting with the antibodies indicated. The experiment was repeated independently 3 times with similar results.

6. The numbers of mice used in in vivo study should be mentioned.

13 mice were used in the A375 MRD model, 6 vehicle and 7 with TEADi (Fig. 8H). This has been added to the figure legend

7. “auto palmitoylation” should be replaced by “autopalmitoylation”.

Fixed, thank you for noting this.

8. The methods for purification of depalmitoylated TEAD protein should be given (or referenced).

A section for protein expression and purification has been included in methods.

Reviewer #2 - Functional genomics (Remarks to the Author):

In this study, Ott et al describe novel TEAD inhibitors that act on resistance to BRAFi inhibitors and block emergence of BRAFi-resistant cells responsible for tumour regrowth.

1. The authors use A375 cells as a model and they have generated SOX10 KO derivatives that then undergo a pseudo-EMT to acquire a more mesenchymal phenotype. The authors should indicate that A375 are MITF-low NCSC like cells. They reproduce many of the effects in MEWO cells with SOX10 KO. MEWO cells are much more melanocytic and express MITF. The fact that SOX10 KO in both leads to a comparable mesenchymal TEAD and TAZ phenotype should be mentioned.

Thank you for the important clarification; we have included this information in lines 87-89 and believe this strengthens the correlation between SOX10 loss and an enrichment of a mesenchymal-like gene signature.

2. Figure 2C is likely to be very difficult to read, can the authors think of a clearer way to present the data. A simple Z-scored heat map would probably suffice.

We have moved the heatmap to Supplemental Fig. 2A. To make the data easier to interpret, we have added a bar plot that displays the top 5 positively enriched gene sets and the top 5 negative enriched gene sets in TAZ knockdown and YAP knockdown samples (Supplemental Fig. 2B). Additionally, we plotted the absolute value of each enrichment score to emphasize that overall, TAZ knockdown leads to a greater degree of transcriptomic perturbation than YAP knockdown (Supplemental Fig. 2C).

3. In the first figures I could find no indication of the expression levels of TEAD1 and TEAD4 or YAP and TAZ between the parental and SOX10 KO A375 cells. The authors mention (line 116) that the underlying mechanism is not known. Immunoblots (showing up-regulation) and immunofluorescence (presence of TEAD and TAZ in the nucleus) would allow the authors to make a stronger statement on this. Blots of parental cells are not shown either in Figure S1.

We answered a similar question from Reviewer 1, Comment 3. We probed for expression of these proteins via western blots of parentals and SOX10 KO (lines 113-117, Supplemental Fig. 1C-D). Furthermore, we performed immunofluorescence in A375 parental and SOX10 KO cells and did not detect a difference in TAZ localization between A375 parental and SOX10 KO cells (Supplemental Fig. 3C). Minor differences in YAP and/or TAZ localization can mediate functional effects and have acknowledged this in lines 198-199. We will continue to examine possible mechanisms in future studies.

4. The authors develop novel inhibitors that clearly block TEAD-TAZ-dependent gene expression. However, there is no mechanistic insight into these effects. The authors claim the compounds target the palmitoylation pocket, but what are the consequences for TEAD proteins in cellulo, are they destabilized, do they no longer interact with TAZ, are they nuclear or cytoplasmic. Additional experiments that address these issues are essential to strengthen the authors hypothesis.

We performed co-crystallization data showing that OPN-9652 covalently interacts with the cysteine 322 residue within the palmitate binding pocket of TEAD1 and have added these new data to Fig. 5D. These data complement the thermal shift data in Supplemental Figure 5A that show *in vitro* data of palmitoylated and depalmitoylated TEAD4 following treatment with OPN-9652 and OPN-9643. These new data confirm that these TEADi bind the palmitoylation pocket.

5. The authors use PD0325901 and PLX4720 as MEKi and BRAFi. However, my understanding is that Dabrafenib and trametinib are the clinically approved and most used combinations. Can the authors comment on this?

We appreciate the reviewer's observation regarding the use of PD0325901 and PLX4720 in our study. While Dabrafenib and Trametinib are clinically approved and commonly used combinations, there are specific considerations for their use in preclinical models. PLX4720 is a formulation specifically designed for use in mice and is analogous to the clinically used Vemurafenib, making it a reliable choice for preclinical studies. Trametinib has known toxicity issues in mice, which can complicate *in vivo* results. PD0325901 is a well-characterized MEKi and is widely used in research to study MEK inhibition. These choices allowed us to minimize toxicity-related complications in long-term (2-3 month) experiments and ensure consistent, interpretable results. We have substantial experience with these drug combinations and believe these compounds are appropriate for our study's goals. Still, we acknowledge the translational relevance of Dabrafenib and Trametinib.

6. In the last section of the results, the authors say that they have previously used PDX models that progress to MRD under MAPKi treatment (lines 292-294). However, they then go on to use A375 CDX as a model for the preclinical validation of the novel inhibitors. Why did the authors not use a PDX model for this critical validation experiment? This referee would very strongly suggest that the TEADi be tested with a PDX model, rather than a CDX model. Alternatively, at least the use of multiple CDX models would give a better idea of the general applicability of the use of these inhibitors. Moreover, the number of mice in each arm is shown neither in the figure, the legend nor the materials and methods. Looking at the Kaplan-Meier in panel 8I one has the impression from the number of events that only a 3-4 mice are in the final cohort.

13 mice were used in the A375 MRD model, 6 treated with vehicle and 7 with TEADi (Fig. 8H). This has been added to the figure legend. We have now included a second CDX model in YUMM1.7 cells showing that TEADi prolongs the onset of resistance following targeted therapy. 10 mice were used in this study, 4 treated with BRAFi + MEKi, and 5 with triple combination. The survival data has been included in Supplemental Fig. 7A. We recognize the value of PDX models in closely mimicking human tumor biology. We did try to grow the TJUMEL41 and TJUMEL54, but their growth properties were slow and precluded their use.

7. Lastly, can the authors benchmark these new inhibitors compared to those previously described and in current clinical trial? What would be the advantage of the new ones over those already characterized in particular IAG933.

OPN-9652 inhibits the TEAD pathway with equal efficacy to the Vivace inhibitor, VT103, and demonstrates greater potency than VT104 via luciferase reporter assays and probing for expression of CTGF and CYR61 via western blot analysis. We answer a similar question from Reviewer 1, comment 4, and the data is shown there. IAG933 has a different binding site and mechanism of action, making direct comparisons with our compounds difficult. We reached out to Novartis but were not able to obtain IAG933.

Reviewer #3 - Melanoma resistance (Remarks to the Author):

This is an interesting, significant and potentially relevant study in the field of cancer resistance, that might lead to future experimental research to target the onset of acquired resistance to targeted therapies in melanoma, in addition of the current clinical trials with TEAD inhibitors for mesothelioma. The manuscript is well written and the methodology is correctly presented.

At present, the manuscript has several limitations that overall reduce the quality of the study. The main one is already mentioned by the authors on line 116: “Mechanistically, how SOX10 loss leads to up-regulation of TEAD targets is not known.” Adding experimental work on this issue will enhance the manuscript.

Another point that must be considered is that the study focuses on SOX10-deficient melanoma cells tolerant/resistant to MAPKi, which is clearly demonstrated here, but that it could represent a minority of the mechanisms leading to tolerance.

From our previous studies (PMID: 35296667), SOX10-deficient melanoma cells represent a major mechanism of drug tolerance. However, mechanisms leading to drug tolerance are still understudied, largely because minimal residual disease (MRD) cannot be easily detected in patients. Investigating any potential pathways contributing to tolerance makes it crucial, as they may all be relevant to understanding the broader phenomenon. Additionally, there is a key distinction between tolerance and resistance: tolerance is an early, often reversible state that can lead to the more permanent and clinically observed resistance seen in patients. Our study focuses on the earlier, more elusive phase of tolerance, which could be foundational in preventing or delaying the onset of resistance. We believe that these findings contribute valuable insights into this critical and less-explored area. Still, we have acknowledged other mechanisms of resistance in our discussion in lines 371-373

Also, the study is performed with only one SOX10-silenced BRAF-mutant melanoma cell line (although two different clones), in addition of the BRAF wt Mewo melanoma cells. It will be important to include an additional BRAF-mutant melanoma cell line to have a more robust output.

To expand the study to include additional cell line models to strengthen the robustness of our findings, we treated drug-tolerant MeWo SOX10 KO cells with TEAD inhibitors to show that TEADi resensitizes the MeWo SOX10 KO cells to Trametinib (Fig. 7B). Furthermore, we have generated stable MeWo cell lines that express inducible TAZ-S89A and YAP-S127A and show that mutant TAZ and YAP are sufficient to induce tolerance to Trametinib in MeWo parental cells (Supplemental Fig. 3D). Additionally, we used a BRAF-mutant

YUMM1.7 xenograft model and show that median survival increases statistically significantly in the triple combination model ($p = 0.0243$, Log-rank test; Supplemental Fig. 7A).

The study lacks experimental data showing YAP/TAZ translocation to nucleus, a step required for TEAD activity. Linked to this point, does SOX10 depletion lead to dephosphorylation of YAP/TAZ in the cells used in this study?

To address these points, we performed immunofluorescence analysis of TAZ localization and found no significant difference in TAZ nuclear localization between SOX10 KO cells and parental controls (Supplemental Fig. 3C). Additionally, we assessed the phosphorylation status of YAP at three key LATS1/2 mediated phosphorylation sites. We observed no changes in phosphorylation between parental and knockout cells (Supplemental Fig. 1C).

Measuring cell growth by analyzing cell culture confluency does not seem the most appropriate method, as some cells in early or advanced apoptosis might be included in this measure.

To address this, we probed for the expression of cell death markers such as gasdermin E cleavage and PARP cleavage and included as new data in Fig. 7C. We also tested cell cycle effects in MeWo SOX10 KO cells through EdU assay and saw no comparable difference in EdU incorporation between Trametinib treated cells and combination therapy (Response Fig. 3). These additional methods provide a more comprehensive cell growth and viability assessment and suggest that reduced confluence detected via Incucyte assays is due to cell death in the combination treatment rather than a reduction in cell cycle progression.

Response Figure 3: Quantification of cell proliferation using an EdU incorporation assay. Bar plot depicting the percentage of EdU-positive cells in MeWo 4.11 following treatment of 50 nm Trametinib, 2 μ M OPN-9652, or combination for 48 hrs. Cells were incubated with 10 μ M EdU for 16 hrs, followed by fixation and Click-iT EdU staining to detect proliferating cells. Data represent the mean \pm SEM from technical triplicates. Statistical significance was determined using One-way Anova, **** $p < 0.001$.

In addition, authors must give more convincing data on the specificity of the TEAD inhibitors, by including additional controls. Authors have limited their TEAD-dependent gene signature to CTGF and CYR61; however, DKK1, WNT5A, TGF-beta2, MHCY and GLUT3 are also TEAD targets.

We strengthened the analysis of TEAD inhibitor specificity and performed RNA sequencing analysis on both A375 parental and SOX10 knockout cells treated with the OPN inhibitors. Following treatment, our results revealed a significant reduction in additional TEAD target genes, including DKK1, TGF-beta, and MYC (Supplemental Fig. 5B). Our RNA-seq data did not show a change in RNA expression in WNT5A or GLUT3 following TEADi. To further validate our inhibitors' specificity, we also compared our RNA-seq data to five published

YAP/TAZ gene signatures. OPN-9652 and OPN-9643 -treated cells showed strong overlap with these established signatures, reinforcing the conclusion that the observed transcriptional changes are indeed mediated by TEAD inhibition (Fig. 6F). We included these comprehensive data in the revised manuscript to provide a more robust analysis of OPN-9652 and OPN-9643 gene modulation.

Moreover, it is known that TAZ overexpression upregulates PD-L1, and YAP/TAZ regulates TGF-beta signaling, adding more complexity to the system. In fact, EGFR- and TGF-beta-dependent signaling are intertwined with YAP/TAZ signaling.

Further exploration of our RNA-seq analysis found a statistically significant positive enrichment of TGF-beta signaling with TAZ knockdown but not YAP1 knockdown (Supplemental Fig. 2A). Additionally, enrichment plots from our analysis show that there is a positive enrichment of TGF-beta signaling following TEADi treatment in A375 SOX10 KO cells, although not statistically significant (Response Fig. 4). These findings underscore the signaling network's complexity and suggest a more nuanced role for TAZ-TEAD in regulating TGF-beta signaling.

Response Figure 4: Enrichment plots showing up-regulation of HALLMARK TGF-beta signaling in A375 SOX10 KO cells treated with OPN-9652 and OPN-9643.

The study will also improve if reported molecular players associated with tolerance are included. For instance, tolerance has been linked to increased NGFR and KDM5B, and to decreased Melan A expression. Furthermore, it will be interesting to analyze MITF, a downstream target of SOX10 which could have some links to the TEAD signaling.

We analyzed the expression of MITF, Melan A, KDM5B, and NGFR in the context of drug tolerance using the single-cell RNA-seq dataset from Rambow et al. on minimal residual disease (MRD). Our analysis revealed that MITF and Melan A were significantly lower in the drug-tolerant invasive cell state, while KDM5B expression was higher, consistent with the reviewer's suggestions. We did not observe a significant difference in NGFR expression in the invasive cell state. These findings are included in the revised manuscript to provide a more comprehensive understanding of the molecular changes associated with SOX10 deficiency and drug tolerance in melanoma in Supplemental Fig. 1A

Specific points

- Fig. 1.

Panel A. Figure legend mentions MeWo cells which do not appear there.

We have added MeWo mesenchymal enrichment plots to Fig. 1A

Panel B. What do the data mean in terms of SOX10 downregulation?

These data show that in the invasive cell state, there is low SOX10 expression and high CTGF and CYR61 expression. SOX10 KO cells are more invasive/mesenchymal and have high TEAD activity compared to their parental cells.

Panel C. Significance is especially strong in Mewo cells, and modest in 2.18 and 4.21 cells.

We agree. Both cell lines have an overall enrichment, and these data were validated via western blot analysis in Fig. 1D.

Panel E. Expression of YAP and TAZ is missing, although appearing on Fig. 3. It will be important to add it here to better link to induction of CTGF and CYR61. These data will also improve by showing 4.21 cells reverses induction of CTGF and CYR61.

We have added YAP1 and TAZ blots to Supplemental Fig. 1C

Panel F: Does SOX10 regulates Merlin? Comment on this.

We did not detect a change in Merlin expression in several resistant cell lines that lost expression of SOX10. See comment in response to Reviewer 2, comment 5, that shows Merlin/NF2 levels are unchanged. These data show that while Merlin loss is observed in the A375 SOX10 KO cells, SOX10 expression does not regulate Merlin expression.

Line 112: A reference is missing.

Reference has been added, thank you

Fig. 2.

Panel A. Text mentions YAP1-knockdown and TAZ-knockdown, but proof of silencing does not show up until Fig. 3A. It will be better to add it here.

We have altered the figure order and text accordingly to show the proof of silencing before RNA-seq data. We have also added a bar blot confirming the depletion of either YAP1 or TAZ in our RNA-seq data (Fig. 2B).

Panel D, step 3. Correct “data” (instead of sata).

Thank you for noting. This has been corrected

Panel E. Which siRNA have been used, 1 or 2?

Both siRNA were used (2 cell lines and 2 siRNAs for rigor).

Line 138. Text mentions. “We observed a greater enrichment in all 4 gene sets following TAZ...”. In fact, data show enrichment in one set, a tendency in another, and no enrichment in the other two.

All four genesets had a greater degree of enrichment following TAZ knockdown than the degree of enrichment in YAP1 knockdown. All conditions are significant in TAZ knockdowns, only about half are significant in YAP1 knockdowns and to a much lesser degree.

Line 139. Changing suggesting for demonstrating will be more appropriate.

Thank you for the correction. The text has been amended

Line 160. Add “A375” in SOX10-deficient cells.

Thank you for the correction. The text has been amended.

Fig. 3.

Panel A. It will be important to show YAP and TAZ expression in parental A375 cells for comparison.

We have included these data in Supplemental Fig. 1C and 1D, along with the expression of TEAD proteins between parental and SOX10 KO cells.

Panel B. As indicated above, it will be important to show the YAP/TAZ translocation to nucleus in cells expressing the active YAP/TAZ forms. Figure legend does not mention how many times was this experiment performed. Can authors also show data on Mewo cells?

This experiment was performed independently three times. We generated additional MeWo cell lines that express YAP S127A and TAZ S127A. These data have been included in Supplemental Fig. 3B. Additionally, we performed IF for HA-TAG in A375 -S127A cells in the absence and presence of DOX (Response Fig. 5). These data show that the active form is expressed and localized in the nucleus.

Response Figure 5. Immunofluorescence images of A375, cells stained for HA-TAG (green) and DAPI (blue). The experiment was performed independently three times, and representative images are shown.

Panel C. As indicated above, it would be better to measure tolerance with another type of growth analysis. Also, parental A375 cells seem not to be affected by BRAFi+MEKi. Can authors explain this fact?

Parental cells in the absence of Dox (left-hand panel Fig. 4A) were treated with either vehicle (solid lines) or BRAFi+MEKi (dotted lines) and are affected by the BRAFi+MEKi treatment. We clarified figure to show solid lines are cells treated with DMSO.

Panels D, E. Data will improve by adding parental A375 for comparison. 4.21 cells display a clear reduction in pErk1/2 upon treatment with BRAFi+MEKi, but their growth seems quite similar to control cells. Any comment? The p values are a bit confusing the way they are shown on panel E.

We used a molecular approach to reduce high TEAD1 activity in SOX10 KO cells. To this end, we used a dominant negative TEAD construct that blocks YAP1/TAZ binding to TEAD. 4.21 SOX10 KO cells are tolerant to BRAFi + MEKi; hence, there is a modest effect but statistically significant reduction in cell growth. We have added lines to clarify the conditions being compared.

Figure 4.

Panel C. Must show SOX10 expression along the lanes.

SOX10 expression remains unchanged following the depletion of YAP and TAZ. This a similar concern addressed from Reviewer #1, Comment 5.

Panel D. See comment on Panel C of Fig. 3.

We appreciate the reviewer's perspective. In this figure, we are showing that the depletion of TAZ affects the confluency of the cells. This figure is neither measuring drug tolerance nor the effects of BRAFi + MEKi.

Fig. 5.

Why/How did they choose these molecules to inhibit the TEAD targets? (only based in the structure of cysteine residues or by any other screening experiment?)

We performed screening with reporter assay, cell growth inhibition assay, protein thermal shift assay and co-crystallization to guide compound selection. These new data have been included in Fig. 5D and Supplemental Fig. 4A-B.

The data will definitely improve if adding a non-inhibiting control from the screens. In lines 208-209 appears compound OPN-9651, but is not longer mentioned or shown in the figure. Is it a negative control?

OPN-9651 is a tool compound that is structurally very similar to OPN-9652 and also functions as a covalent TEAD inhibitor. We have not tested OPN-9651 and have, therefore, removed the text referencing this inhibitor. We have compared VT106, a 50x less potent enantiomer of VT104 and have compared the reduction in luciferase expression following treatment to other VT compounds Response Fig. 6).

Response Figure 6: A375 crSOX10 #2.18, and A735 crSOX10 #4.21 cells were treated with 2 μ M of either VT103, VT107, or VT106 for 24 hrs and then cells were lysed. Firefly luciferase activity was measured via Dual-Luciferase[®] Reporter Assay System. The experiment was repeated independently three times with similar results. ****p<0.0001, One-way ANOVA.

Line 212. Missing a reference.

Thank you. The reference has been added.

Lines 212-215. Data on dose response not shown. Only results in Fig. 5C. Could authors show it in a supplemental figure?

Thank you for this suggestion. We have included the growth data and the reporter data from where the IC₅₀ values were generated and included the graphs in Supplemental Figure 4A-B.

Fig. 6.

Panel A. Separation of data in three panels does not allow to visualize whether increased luciferase activity especially in 2.18 cells is statistically significant compared to parental cells in the absence of both inhibitors.

Because these are stably transduced cell lines, we cannot make assumptions of significance based on luciferase transduction efficiency. We have removed the statement that we observe a higher luciferase signal in the SOX10 KO cells.

Panels A-C: Show data also using an inactive TEAD inhibitor as control

While we agree the data would improve by using an inactive control compound with similar structural components to the two novel compounds used in this study, we do not have access to an inactive control. Therefore, we used the vehicle (DMSO) as a control. As a complement, we performed experiments with the Vivace series of compounds and included an enantiomer control in these experiments. These results show that the OPNA inhibitors show similar levels of TEAD reporter inhibition to the VT inhibitors and that an enantiomer control (VT106) is ineffective (Fig. 6, above).

Panels B, C: If possible, show expression of a non-TEAD target (in addition of the HSP90 loading control).

We have included actin expression in Panel B, which remains unchanged from the TEADi treatment.

Panel D. p value? One replicate? Concentration of inhibitors of 2 mM?

This experiment was done in technical triplicates and inhibitors were both treated at 2 μ M. We have amended the figure legend to reflect this.

Panel E. In order to show specificity, and to increase the observation of the direct relationships between the observed alteration and TEAD inhibition, can you compete TEAD inhibitor with expression of active TAZ to check reversion (or reduction) of some of the targets?

We have performed this experiment and included it in Supplemental Figure 4F.

Fig. 7. Panel A. Lacking parental A375 cells

The comparison we aimed to emphasize was between BRAFi +MEKi and the triple combination. Growth data comparing A375 parental and A375 SOX10 KO in the presence of BRAFi +MEKi has been published in Caparelli et al. 2022 PMID: 35296667

Panel C. Is OPN-9643 exerting similar effects of the one displayed by OPN-9652?

Only OPN-9652 was tested in 3D models.

Panel F. Missing control Dox alone

In this experiment, we wanted to compete active TAZ with TEADi to determine whether TEADi is sufficient to restore sensitivity to this induced resistant model. The comparison between Dox alone and Dox + BRAFi+MEKi is shown in Fig. 4A.

Fig. 8.

Panel B. Enrichment plot for PBRT resistant cells (from 1205Lu treated with BRAFi/MEKi)?

We have included the enrichment plots for the PBRT cells treated with PLX8394, a paradox breaker BRAFi (Fig. 8A).

Panels C-E. Different times used for the analyses. Any comment?

The times used are consistent with other experiments in this paper. In panel C, cells are untreated, so 24 hour collection is sufficient. In panel B, all siRNA used in this study were collected at the 72 hour time point.

Panel D. siTAZ2 seems to also silence YAP, and siYAP2 also silences TAZ? Any explanation?

We aligned the TAZ siRNA #2 sequence with YAP and there was a 9 nucleotide overlap, which is likely insufficient to cross-react with YAP. We rule out feedback loops since this phenomenon is only observed in one siRNA sequence. It is possible that siRNA-induced silencing will generate non-specific effects, such as general responses to the presence of dsRNA molecules, responses to transfection, and responses to experimental manipulation of the cells. In this event, it may be one or a combination of non-specific events. This is why we used two siRNAs.

Panel F. Immunoblotting shows data from total tumor. The possible contribution of the tumor microenvironment must be commented. Also, can you also show data on CTGF?

We were unable to blot for CTGF expression in our xenograft tissue. However, we validated our CYR61 data with a second antibody (CST (#14479S), which showed consistent levels of expression to data in Fig. 8F (Response Fig. 7).

**Response Figure 7**. 4.21 SOX10 KO cells were injected into mice. Once tumors reached 100 mm$^3$, mice received either OPN-9652 (50 mg/kg) for 3 days or vehicle. Tumors were harvested and analyzed by Western blotting with the antibodies indicated.

REVIEWER COMMENTS

We thank the reviewers for their thoughtful and constructive feedback, which has significantly strengthened our manuscript. A central aim of this study is to characterize two novel TEAD inhibitors, which was further supported in the previous revision through the incorporation of co-crystallization data demonstrating compound binding to TEAD.

In response to reviewer comments, we have also expanded our mechanistic insights by exploring the established relationship between the AP-1 complex and TEAD transcription factors (PMID: 25865119, PMID: 26832411 PMID: 26258633). AP-1 proteins are known to interact with TEAD to regulate target gene expression. Specifically, we show that SOX10 knockout (KO) melanoma cells exhibit upregulation of the AP1 complex protein, c-Jun. We also show that targeting c-Jun reduces the expression of TEAD target genes, CTGF and CYR61. New data in Supplemental Fig. 2B-2C show that while SOX10 loss modulates YAP/TAZ-TEAD target genes without altering YAP or TAZ expression, its loss is associated with chromatin remodeling leading to increased accessibility of TEAD sites (PMID: 32753671). These new data provide insights to the mechanism of action and therapeutic relevance of TEAD inhibition in melanoma.

Reviewer #1 (Remarks to the Author):

The authors have greatly improved the manuscript with the revision. All of my previous comments have been addressed. I support the publication of the paper.

Reviewer #2 (Remarks to the Author):

Given that the authors state that there are no changes in TEAD/TAZ expression/localization upon SOX10 KO, the question of the cross-talk between these pathways becomes a major issue that has not really been addressed.

Based on prior studies showing crosstalk between TEAD and AP-1 complexes (PMID: 26832411, PMID: 26258633, PMID: 27802171), we examined AP-1 protein expression in SOX10 knockout cells. We observed increased c-Jun expression in SOX10 knockout cells by RNA-seq and Western blot. Importantly, we demonstrated that c-Jun knockdown reduces expression of the TEAD targets, CYR61 and CTGF, providing a mechanistic link between SOX10-deficiency, c-Jun and TEAD pathway, despite no overt change in TEAD/TAZ expression or nuclear localization. Together, these findings suggest that increased c-Jun expression in SOX10 knockout cells enhances TEAD transcriptional activity.

While we now include c-Jun data, we would like to emphasize that we have explored multiple angles to identify how SOX10 loss leads to upregulation of TEAD activity. For example, recent work has identified PKN2 as an activator of YAP/TAZ-TEAD signaling, and a driver of the mesenchymal-like cell state (PMID: 39560431). Despite the increased TEAD activity and mesenchymal-like cell state of SOX10 KO cells, we did not detect changes in PKN2 expression between SOX10 KO cells and parentals. Other work shows that SOX9 promotes the nuclear translocation of YAP, potentially driving YAP-TEAD signaling without a change in expression of YAP (PMID: 38653754). To test the potential role of SOX9, we overexpressed SOX9 in SOX10 +ve/SOX9-ve parental cells, but expression of CTGF and CYR61 remain unchanged. Thirdly, we explored VGLL proteins, that compete with YAP and TAZ for TEAD binding. While we would expect VGLL proteins to be reduced in SOX10 KO cells to allow for increased YAP/TAZ-TEAD complexes, our RNA-seq data revealed that

VGLL3 is increased in MeWo SOX10 KO cells, underscoring the complexity of feedback and feedforward loops that are involved in Hippo signaling.

What is the effect for example of TEAD or TAZ silencing in the SOX10-expressing parental lines?

In lines 160-164, we state: Principal component analysis (PCA) did not separate YAP1 knockdown and TAZ knockdown in A375 parental cells from control A375 cells, likely since the downstream activity is already low in the parental cells (Fig. 2C). These data indicate that YAP1 and TAZ depletion do not elicit major transcriptomic changes in SOX10-expressing cells.

We have provided additional analysis here, that show in SOX10-expressing parental A375 cells, knockdown of either YAP1 or TAZ did not result in significant changes in cell growth, consistent with the low basal activity of the YAP1/TAZ-TEAD axis in this context. (Response Fig. 1A). We performed RNA-sequencing of A375 parental and SOX10 KO cells following knockdown of either YAP1 or TAZ. The full data set including knockdown of YAP1 and TAZ in A375 parental and SOX10 cells is publicly available and can be found under GEO accession number GSE259388. Our analysis indicated that while in SOX10 KO cells, TAZ knockdown led to the significant upregulation of 22 gene sets, only 5 were upregulated in parental cells. Additionally, 9 gene sets were significantly downregulated in SOX10 KO cells following TAZ silencing, whereas none were downregulated in the parental context (Response Fig. 1B). These results support the conclusion that YAP/TAZ-TEAD signaling plays a more prominent role in the SOX10-low, drug-tolerant cell state, compared to their parental counterpart. (SOX10-proficient)

Response Figure 1: A. A375 Parental cells were treated with reagent alone (no siRNA), non-targeting control siRNA, or siRNAs to either YAP1 or TAZ. Cells were imaged using IncuCyte Live Cell

Analysis System. Shown is the mean \pm SEM percent plate coverage from three independent experiments. **B.** Venn diagram showing total number of gene sets significantly enriched following siTAZ knockdowns in parental cells and SOX10 KO cells.

What signaling pathways are changed upon SOX10 KO?

We have previously published (PMID: 35296667, Fig. 2) that by transcriptional analysis, SOX10 knockout alters pathways such as TGF β signaling and TNF α signaling via NF κ B. The RNA-seq data is publicly available through NCBI BioProject numbers PRJNA701949 and PRJNA748713. We now describe this detail in the revised manuscript (lines 416-417).

Does the phosphorylation state of TAZ or TEAD change?

We tried to assess the phosphorylation state of TAZ but found that the antibody (phosphoSer89-TAZ, E1X9C) is discontinued. Little is known about the mechanism and regulation of TEAD phosphorylation. While TEADs 1-4 can be phosphorylated by protein kinase A and protein kinase C

(PMID: 10931933, PMID: 11313339), phospho-TEAD antibodies are not available limiting our ability to address phosphorylation state changes in TEADs.

Once activated, can SOX10 re-expression repress the activity of the TEAD/TAZ pathway?

Yes, re-expression of SOX10 in SOX10-deficient cells represses expression of CYR61 and ANKRD1 in MeWo SOX10 KO cells (Response Fig. 2). These data are part of an ongoing work to map the regions of SOX10 with phosphorylation and sumoylation mutants and, hence, are not included within this manuscript.

Response Figure 2: MeWo parental, MeWo 4.11 SOX10 KO, MeWo 4.11 SOX10 KO-LacZ, and MeWo 4.11 SOX10 KO re-expressing SOX10, cell lysates were analyzed by Western blotting with the indicated antibodies.

Another minor outstanding issue is the very small number of mice used for the CDX experiments (6 and 7 in one group and 4-5 in the other). This really is at the limit of statistical significance, perhaps the authors could add a supplemental Figure showing the growth of each individual tumour in each arm as well as the aggregate shown in the main figure.

We have now included individual tumor curves for A375 xenografts (Supplemental Fig. 7A) and YUMM1.7 xenografts (Supplemental Fig. 7D).

Reviewer #3 (Remarks to the Author):

Through the text, authors use the terms tolerance and growth without a clear distinction at given experimental contexts. This must be clarified.

We have revised the manuscript text in the section starting on lines 239 to more clearly describe the functional effects of TAZ during BRAFi + MEKi treatment. Specifically, we now explicitly state that TAZ enhances cell growth in BRAFi + MEKi treatment conditions, which more accurately reflects the observed effects.

Discussion lacks a conceptual approach of what the results of this study provide for treatment of melanoma cells in the context of low/absence of Sox10.

To address this point, we have added text to the discussion that emphasizes the therapeutic implications of our findings in the context of SOX10-low, drug tolerant melanoma cells. Specifically in lines 477-482, we write,

“Together, our findings support a model in which SOX10 loss promotes a shift toward a drug-tolerant melanoma cell state characterized by increased reliance on TEAD-driven transcription and sensitivity to TEAD inhibition, indicating a context-dependent vulnerability. Our findings highlight a potential clinical strategy in which standard-of-care therapy is used initially to reduce tumor burden, followed by addition of a TEAD inhibitor at the minimal residual disease stage, which is designed to eliminate drug-tolerant persister cells.”

Specific points

- The previous revision mentioned that this study will be strengthened by inclusion of data with at least an additional BRAF-mutant melanoma cell line silenced for SOX10. The authors have used the mouse melanoma cell line YUMM 1.7 (BRAF-mutant) directly in xenograft studies using the triple inhibitor combination as shown in the final part of the manuscript. Although they show that

median survival increases statistically significantly with this combination, data is inconclusive, as they are missing the links between SOX10 absence, YAP/TAZ and TEAD. This must be addressed.

To address this point, we now include western blot data showing that YUMM1.7 cells are SOX10-low cells that express canonical TEAD markers indicating that the YAP/TAZ-TEAD pathway is activated in this cell line (Supplemental Fig. 7B). These data provide the link between SOX10 deficiency with TEAD activity in the YUMM1.7 cell model.

- Another comment from the first revision related to the lack of experimental evidence regarding YAP/TAZ translocation to nucleus, a step required for TEAD activity. The data also remains inconclusive, as TAZ and YAP should translocate to nucleus once they are dephosphorylated to activate TEAD.

Supplemental Fig. 4C shows that TAZ localizes in the nucleus in both A375 and SOX10 KO cells. While small changes in nuclear localization that cannot be detected via IF may result in large downstream transcriptomic alterations, we analyzed other possible mechanisms. We add new data showing up-regulation of c-Jun in SOX10-deficient cells and the contribution of c-Jun to regulation of TEAD targets (Supplemental Fig. 2). Additionally, we explore publicly available data that shows that SOX10 loss leads to chromatin remodeling, and that there is an increase of DNA accessibility at *CYR61* and *CTGF*. This data provides insights about how SOX10 deficiency led to YAP/TAZ/TEAD pathway activation, despite no changes are observed in the nuclear localization.

- To measure cell growth by analyzing cell culture confluency.

Although authors have addressed the measurement of cell growth and apoptosis in the context of using the OPN inhibitors with MeWo SOX10 lo (new fig. 7C), this reviewer still considers that looking at cell confluency to define drug-tolerant persisters might not take into account issues such as cell shape and spreading, and cell-cell packing.

We agree that factors such as cell shape and spreading can influence confluency measurements. To address this concern, we extracted representative images directly from the IncuCyte to confirm that the graph reflects dramatic differences in confluency (Supplemental Fig. 7C). Furthermore, our approach is consistent with methods previously employed by other labs (e.g. Oren et al. Nature PMID: 34381210), in which live-cell imaging was used to visualize drug-tolerant colonies over time. This citation has been added. The images support that the reduction in confluency in response to the triple combination and combined with reduction of cell cycle proteins (Fig. 6D) and increased apoptosis markers (Fig. 7C) show that the effects are due to decreased cell growth and/or increased apoptosis rather than artifacts related to spreading or packing.

- Analysis of the expression of molecular players associated with tolerance, including NGFR, KDM5B and Melan A. Also checking the levels of MITF, a downstream target of SOX10 which could have some links to the TEAD signaling. Authors show in Supplemental Fig. 1A the expression of these markers in the context of drug tolerance using the single-cell RNA-seq dataset from Rambow et al. on minimal residual disease. Although interesting, they have not tested the levels of these markers in their SOX10 KD cells, which will make the data more convincing.

As the Reviewer requests, we further test the levels of markers in SOX10 knockout cells. A375 cells are MITF-low and have been previously described as NCSC-like cells (PMID: 3275367); however, MeWo cells express MITF and our lab has previously published that MeWo SOX10 KO cells express lower levels of MITF (PMID: 35296667, Fig. 3A). We mention this in the manuscript lines 96 and 108. We include MelanA and NGFR data as Supplemental Fig. 1B, lines 107-112). KDM5B levels did not consistently change, and the data are provided for the reviewer in Response Fig. 3.

Response Figure 3: A375 parental, A375 2.18 SOX10 KO, A375 4.21 SOX10 KO cells and MeWo parental, MeWo 2.1 SOX10 KO, and MeWo 4.11 SOX10 KO cell lysates were analyzed by Western blotting with the indicated antibodies. The experiment was repeated independently two times with similar results.

- Previous line 160.

As indicated by authors, text has been amended to ...”suggest that TAZ is the major co-activator of the TEAD transcriptome in cutaneous melanoma cells”....

I would suggest to change the last part of the sentence to “...of the TEAD transcriptome in SOX10 low cutaneous melanoma cells”

We agree with the reviewer and have made this change (now lines 199).

- Comment on old Fig. 3B. Translocation of YAP/TAZ to nucleus.

..author’s data conclude (subheading, line 121) that “TAZ is a major regulator of TEAD targets in SOX10-deficient melanoma cells.” Therefore, it will be important to show the new data of activated TAZ (S89A mutant) in a figure, not in a Response figure.

We agree and have moved the data to Fig. 3G and added text to lines 228-229.

Furthermore, in the blot of the new Supplemental Fig. 3B they show HA-tag expression in TAZ S89A but not in YAP S127A. This is contradictory to what authors display in response Fig. 5. Also, on this old Fig. 3B (now new fig. 3F), there is no HA label on the blot corresponding to the YAP S127A cells. Reason?

We apologize for any confusion. The YAP S127A construct used in these experiments does not contain an HA tag; hence, we used a total YAP1 antibody to detect its expression. We have added clarifying text within the manuscript (lines 228-229).

- Comment on old Fig. 3D, 3E. Unless I missed in the new version, I can find comparison with parental A375 cells.

We didn't make the inducible dominant negative TEAD1 in the parental cells since we were focused on inhibition of the pathway in SOX10-deficient cells that exhibited high levels of TEAD1 activity. In our opinion, it is a hard comparison to make since the levels of induction in distinct cell lines would have to be the same.

- Comment on old Fig. 4C.

The data is now shown in Fig. 3D, which displays SOX10 expression in WM983B cells independently of high levels of the YAP/TAZ/TEAD targets CTGF and CYR61. The authors mention in the revised version, lines 177-180: ...”There was no correlation between SOX10 expression and predicted YAP1 or TAZ dependency; however, there was a correlation between SOX10 expression and TEAD1 dependency (Supplemental Fig. 3A) and between predicted TAZ dependency and predicted TEAD1

dependency (Fig. 3C).” Yet, data on Figs. 1 and 2 with SOX10 KD A375 cells suggest a pathway linking SOX10 with YAP/TAZ and TEAD. Later, in lines 199-202, authors mention: ...”Despite there being no correlation between SOX10 expression and YAP1 or TAZ dependency at steady-state, further analysis is warranted to determine mechanisms that TAZ drives a SOX10-deficient drug-tolerant phenotype.” What authors mean for steady-state? A clear explanation for these discrepancies must be provided.

We used "steady-state" to refer to the baseline, drug-naïve condition of the cells analyzed in the DepMap data. Analysis of DepMap CRISPR dependency data shows that only 5 out of 62 melanoma cell lines exhibit dependency on TAZ, and none are dependent on YAP1. This supports the conclusion that TAZ, rather than YAP1, is the predominant transcriptional coactivator of TEAD in melanoma and the majority of melanomas are not dependent on YAP1/TAZ signaling at basal conditions. In the DepMap cell lines, there was no correlation between TAZ dependency and SOX10 expression. Consistent with majority of melanoma cell lines not being dependent on YAP1/TAZ signaling. In line with these findings, depletion of TAZ (Fig. 4B) or treatment with a TEADi (Supplemental Fig. 7A) does not significantly impact cell growth of SOX10 KO cells in the absence of BRAFi + MEKi. We have removed the predictive data linking SOX10 expression and TEAD1 dependency since it is not supported by functional testing.

By contrast, when SOX10 KO cells are exposed to BRAFi + MEKi, TAZ depletion significantly impairs cell growth (Fig. 4B). Our findings suggest that TAZ-TEAD signaling is transcriptionally activated in the absence of SOX10, and it becomes functionally relevant following BRAFi + MEKi treatment (Fig. 4B and Fig. 7A-B). This is consistent with other publications showing that TEADi do not affect the cell growth of MAPKi-resistant cells, but resensitize resistant cells to MAPKi (PMID: 37277530). We have revised the manuscript to more clearly define "steady-state" (now noted as "drug-naïve state"). Our main conclusion of this section is that TAZ rather than YAP1 is the main co-activator utilized in melanoma cells.

- Comment on old Fig. 5.

In the response Fig. 6, authors show data using several TEAD inhibitors of the VT group. They mention that VT106, a 50x less potent enantiomer of VT104, displays a lower reduction in luciferase expression. How is this inhibition in comparison with the OPN inhibitors in the same, not separated experiments? I understand that sometimes is difficult to find the appropriate control, but the above mentioned comparison might provide more useful information.

We appreciate the reviewer’s suggestion. VT106, as a 50x less potent enantiomer of VT107, serves specifically as an internal control for the VT inhibitor series, which are non-covalent inhibitors, different from the covalent OPNA inhibitors. As the reviewer requests, we have performed a luciferase reporter assay to evaluate the level of TEAD inhibition by each of the OPNA compounds, including the comparison of VT107 and its control compound, VT106. We have included these data as Supplemental Figure 6A and text in lines 323-326.

In the previous version of the manuscript, the IC₅₀ for NCI-H226 was 3 uM in Fig. 5C. Now it shows 100 nM. Please, explain this change. Also in this panel, the IC₅₀ MSTO-211 TEAD values do not match to what is written in the text.

We apologize and should have explained before. In the original submission, the IC₅₀ values provided were averages derived from initial experiments conducted at Plexikon; however, the original dose-response curves requested by the reviewer were no longer available after the company transitioned to OPNA-BIO. Hence, we performed new experiments to generate the data

presented in the current version of Fig. 5C. To ensure accuracy, the current IC50 values are based on at least three independent experiments, each performed in duplicate. We have also updated the text to reflect these changes and corrected the IC50 values for MSTO-211H TEAD to match the figure. We apologize for the confusion and appreciate the opportunity to clarify this point.

- New Fig. 6A.

Points in the A375 4.21 data bars are missing.

Data bars have been added.

Reviewer #3

Authors have addressed the new comments raised by this reviewer. The manuscript now shows a clear improvement, especially on data from Figs. 5-8 and associated supplemental figures, which constitute an interesting characterization of the TEAD inhibitors. However, there are still few pending points that need to be addressed. A main pending concern is that induction of TEAD targets seems not to be always associated with loss or reduction of Sox10.

Specific points

- Fig. 1D: If TAZ is similarly expressed in parental and Sox10-depleted cells (see suppl. Fig. 1D), why there is no induction of CTGF and CYR61 expression in parental cells? Is it possible that the presence of Sox10 blocks the induction?

While we do not observe direct SOX10 binding at the CTGF or CYR61 promoters, it is likely that SOX10 indirectly suppresses their expression through multiple mechanisms. These include loss of the Hippo pathway regulator NF2/Merlin (Fig. 1E), increased chromatin accessibility (Supplementary Fig. 2C), and upregulation of AP-1 proteins such as c-Jun, which enhance TEAD-mediated transcription (Supplementary Fig. 2D–F). Together, these changes following SOX10 loss contribute to the induction of TAZ target genes despite similar TAZ protein levels.

- Suppl. Fig. 1D, E: Authors mention a modest increase in YAP, TEAD1 and panTEAD (lines 127-130). However, the western blots do show remarkable increases. Have authors quantified the blots? Authors must consider to show data from this supplementary figure as part of main Fig. 1, to have data in a more logical and ordered form. Otherwise, the data looks a bit disorganized.

In this same figure (panel D), phosphorylation status of YAP1 and TAZ is shown, but data is not commented in the text.

To better reflect the magnitude of change, we have removed the word “modest” and updated the text to note that YAP phosphorylation increases in SOX10 KO cells, likely due to the increase in total YAP expression (lines 114-118).

- Sentences within lines 146-150 show some repetition. Must clarify

Reduced repetition, now lines 133-136. “Overall, these data show that despite no overt change in TEAD/TAZ expression, SOX10-low, drug-tolerant cells exhibit up-regulation of YAP1/TAZ-TEAD signaling in melanoma, likely through multiple mechanisms including chromatin remodeling and c-Jun upregulation.”

- Line 173 should read: ...melanoma cells depleted of Sox10 for mediating...

Added “depleted of SOX10” (line 159 in revised document)

- Fig. 3D: WM983B cells express Sox10, TAZ, YAP1 and also CTGF and CYR61. Therefore, these cells should have enhanced TEAD-dependent signaling in the presence of Sox10. This is different from data shown on fig. 1D.

WM983B cells are a validation of one of the “TAZ-dependent” cell lines based on DepMap data in Fig. 3A. We confirm in Fig. 3D and Fig. 3E that WM983B are dependent on TAZ signalling for both CTGF and CYR61 expression, and for growth based on incucyte data. The mechanism for up-regulation of TAZ-TEAD signalling in WM983D is unclear at this time.

- Line 219: This new sentence should read:...primary TEAD co-activator in Sox10-depleted melanoma, its requirement...

TAZ functions as the primary TEAD co-activator not only in SOX10-depleted cells but also in models where SOX10 is still expressed. Specifically, we observe TAZ dependency in WM983B

cells, which retain SOX10 expression, and in SOX10-expressing parental cells with TAZ overexpression.

- Fig. 3F: How do you rule out that there is not too much effect of mutant YAP1 due to the fact that it might not translocate to nucleus? Authors show no data on YAP translocation to nucleus. Text related to this figure does not mention the YAP1 data shown in the figure.

The S127A mutation is well characterized and is a known site of LATS dependent phosphorylation (PMID: 20048001, PMID: 24101154). We show an increase in total YAP1 expression in Fig. 3F following dox induction. We did not pursue the analysis of YAP S127A nuclear localization since other have published on this (PMID: 20439427, PMID: 1797491, PMID: 22863277) and we were focused on TAZ effects.

- Lines 250-253: As there are no differences in TAZ expression between parental A375 and Sox10-knockdown cells, would TAZ depletion in parental A375 cells also lead to decreased growth? If so, this would suggest no correlation between TAZ and Sox10.

TAZ depletion does not reduce growth in parental or SOX10 KO A375 cells without BRAFi + MEKi. Rather TAZ knockdown impairs growth in the drug-tolerant SOX10 KO cells, linking TAZ function to the SOX10-deficient tolerant state.

- Fig. 4D, lines 258-259: Unless I am confused, there are no differences in percent confluency between induced and non-induced TAZ-S89A. Therefore, the sentence is unclear.

The purpose of this experiment was to test whether TAZ-S89A-mediated growth under BRAFi + MEKi is dependent on TEADs. When all four TEAD paralogs were knocked down, we observed no difference in percent confluency between TAZ-S89A-induced and non-induced cells. This indicates that the growth-promoting effects of TAZ-S89A are dependent on TEAD expression. Without TEADs, TAZ-S89A is unable to promote drug tolerance. We have revised the text to clarify this conclusion (lines 243-246).

- Fig. 6A, lines 310-313: Data suggest that inhibition is independent of the presence or absence of Sox10

While TEAD inhibitors reduce TEAD driven luciferase activity in the presence of SOX10, TEAD signalling is substantially higher in the SOX10-deficient cells. IN SOX10 expressing parental cells, basal TEAD activity is detectable via sensitive assays such as luciferase reporters and the OPNA inhibitors reduce TEAD-driven luciferase activity.

- Suppl. Fig. 5F, lines 315-317: Again, independently of Sox10

Correct, independently of SOX10.

- New text in lines 324-327: There seems to be no conclusion from data in suppl. Fig. 6A. Are VT106 and VT107 also good TEAD inhibitors? Also in the Rebuttal letter, on Comment on old Fig. 5, there appear to be a confusion between VT104 and VT107.

VT107 (and its less potent enantiomer VT106) are distinct TEAD inhibitors with different mechanisms of action (non-covalent vs covalent) and differing chemical structures from the novel OPNA inhibitors we have characterized here in our manuscript. Their inclusion in the supplementary data was to provide additional context at the reviewer's request. We found a typo in the previous response to reviewers. VT106 is an enantiomer of VT107. These inhibitors have been published and information regarding their structure and characterization can be found here PMID: 33850002.

- Suppl. Fig. 9A-D: On one hand, authors use the Sox10-expressing A375 in NSG models. On

the other hand, panels B-D display data using Sox10-deficient cells. As data show similar results, it is suggested that effects are independent of the presence or absence of Sox10.

Although A375 cells initially express SOX10, it is well established, and we have confirmed in our model (PMID: 31270153) that treatment with BRAFi + MEKi leads to the loss of SOX10 expression as cells enter a drug-tolerant state. Therefore, by the time minimal residual disease (MRD) is established in vivo, the A375 xenografts have lost SOX10 expression, mirroring the adaptive transition observed in patient tumors.

- Line 429, Discussion: I think using “Mechanistically” here is going too far. Authors most soften this sentence.

We deleted 'mechanistically'.